# A neuroprotective tetrapeptide for treatment of acute traumatic brain injury

Aman P Mann [ID][1,6✉], Sazid Hussain[1,6], Pablo Scodeller [ID][2], Hope N B Moore[3,4], Elan Sherazee[3,4], Rachel M Russo[3,4] & Erkki Ruoslahti [ID][1,5✉]

## Abstract

Traumatic brain injury (TBI) is a major clinical problem because of the high incidence and the severity of the subsequent sequelae. Despite extensive efforts, there are no therapeutic drugs clinically approved for treating acute TBI patients. To address this unmet need, we assessed the activity of the tetrapeptide, CAQK, in mice. When administered intravenously shortly after moderate or severe TBI, CAQK accumulates in the injured brain in mice and pigs. CAQK binds to an extracellular matrix glycoprotein complex that is upregulated in injured brain. Treatment of TBI mice with CAQK resulted in reduction in the size of the injury compared to control mice. There was reduced upregulation of the glycoprotein complex, less apoptosis, and lower expression of inflammatory markers in the injured area, indicating that CAQK alleviates neuroinflammation and the ensuing secondary injury. CAQK treatment also improved functional deficit in TBI mice, with no overt toxicity. Our findings suggest that CAQK may have therapeutic applications in TBI.

**Keywords** Traumatic Brain Injury; Neurotherapeutic; Peptide; Brain Injury
**Subject Category** Neuroscience

## Introduction

Traumatic brain injury (TBI) is a global public-health problem because of its prevalence and long-term sequelae associated with it. Each year, ~70 million individuals worldwide sustain a TBI, which is a leading cause of injury-related death and disability (Dewan et al, 2019). TBI survivors often suffer from long-term physical disabilities and cognitive disorders, including depression, drug and alcohol abuse, and increased risk of suicide (Jorge et al, 2004). Current critical care management of TBI focuses on general intensive care support and specific neurocritical care interventions to minimize further damage to the brain by preventing hypoxia, hypotension and hypoventilation. However, there are no clinically

proven effective pharmacologic agents to limit secondary injury or enhance repair (Carney et al, 2017). Furthermore, advanced imaging and monitoring techniques are not available in smaller trauma care centers, making clinical management of patients with severe TBI even more difficult. Therefore, novel therapeutic approaches are needed to improve clinical outcomes for TBI patients.

TBI is heterogeneous and varies widely in severity, clinical presentation, and pathophysiology (Ng and Lee, 2019). Multiple cascades of signaling events, both acute and delayed, are involved in neuronal cell death following a TBI and play a role in secondary damage. Some of these include blood–brain-barrier disruption, neuroinflammation, oxidative stress, and demyelination. Due to this multifaceted nature of TBI, future therapies to treat TBI patients will likely need a pluripotent mechanism of action to successfully modulate the secondary injury cascade, and, in doing so, maximize the likelihood of a successful clinical outcome. Homing peptides can deliver a variety of therapeutic payloads in high concentrations directly to the site of injury, minimizing systemic side effects, and offering an option to facilitate such treatments.

An unbiased screening approach utilizing in vivo phage display in a mouse model of unilateral penetrating brain injury revealed a tetrapeptide, CAQK (cysteine-alanine-glutamine-lysine), that showed specific localization into brain injury from intravenous (i.v.) administration (Mann et al, 2016). Initial data indicated that the peptide extravasates through compromised blood–brain-barrier associated with moderate and severe TBI and does not accumulate in normal brains. In the injured brain, the peptide targets an extracellular matrix (ECM) glycoprotein complex, which is produced in increased amounts and may also be structurally altered in TBI (George and Geller, 2018). CAQK binding to injured regions from human brain tissue was also shown (Mann et al, 2016). Subsequent studies demonstrated that the injury-targeting capability of CAQK was useful for site-specific delivery of different types of payloads to brain and spinal cord injuries (Wang et al, 2022; Wang et al, 2020; Wang et al, 2018b). Other disease-specific homing peptides discovered using similar phage library screening have shown inherent biological activity in the absence of a payload that can be therapeutic (She et al, 2016; Sugahara et al, 2015). We used a pre-clinical controlled cortical impact (CCI) model of TBI to determine whether CAQK might have such an intrinsic activity.

[1]AivoCode Inc., San Diego, CA, USA. [2]Institute for Advanced Chemistry of Catalonia IQAC-CSIC, Barcelona, Spain. [3]Clinical Investigation facility, Travis Air Force Base, Fairfield, CA, USA. [4]Department of Surgery, University of California Davis, Davis, CA, USA. [5]Sanford Burnham Prebys Medical Discovery Institute, La Jolla, CA, USA. [6]These authors contributed equally: Aman P Mann, Sazid Hussain. ✉E-mail: amann@aivocode.com; ruoslahti@sbpdiscovery.org

# Results

## CAQK binds tenascin-C in the ECM of the injured brain

We have previously identified a tetrapeptide, CAQK, that accumulates in the ECM of the injured brain. Other homing peptides, in addition to accumulating at their target, have been found to have inherent functional effects at the target tissue (Maldonado et al, 2023; Sugahara et al, 2015; She et al, 2016). We reasoned that knowing the target molecule CAQK binds to would likely give hints of the possible nature of such an activity. Previous work has shown that CAQK binds to a chondroitin sulfate proteoglycan (CSPG)-rich protein complex, the expression of which is elevated following an injury (Mann et al, 2016). We confirmed this result by testing the binding of CAQK to CSPG complex isolated from chicken brain. Fluorescence polarization assay, which is widely used to measure interactions of small fluorescent ligands with larger binding partners (Lea and Simeonov, 2011), showed dose-dependent binding of fluorescein (FAM)-CAQK to the CSPG complex, not seen with a control peptide (Fig. 1A). Pretreatment of CSPG with chondroitinase ABC to remove chondroitin sulfate and dermatan sulfate side chains of proteoglycans had no effect on the FAM-CAQK binding (Fig. EV1A). This suggests that CAQK interacts with a protein, not a glycosaminoglycan, in the CSPG complex. To identify a cell line that could be used as a source of the CSPG complex, we tested the human glioblastoma cell line U251. FAM-CAQK bound to proteins in the supernatant culture media of the U251 cells and to CSPGs complex purified from these cells (Fig. EV1B). Proteomics analysis showed that tenascin-C (TnC) was present in both the U251 and chicken brain CSPG complexes (Fig. EV1C). Indeed, CAQK efficiently and dose-dependently bound to TnC purified from U251 cells in the fluorescent polarization assay (Fig. 1B). Only marginal binding of CAQK to other related glycoproteins, including tenascin-R, was observed. The cysteine in the peptide was required for the binding, as the AAQK peptide or a cyclic, disulfide-bonded peptide containing the CAQK sequence did not show any binding (Fig. EV1D). Furthermore, the addition of glutathione or iodoacetamide (to block the free sulfhydryl in CAQK) completely abrogated the binding of CAQK peptide to TnC (Fig. EV1E). Previously, we have shown that i.v. injected CAQK accumulates in the injured side of the brain and not in the contralateral side (Mann et al, 2016). No other organs showed peptide accumulation except the kidneys, the main site of clearance of small peptides. We also showed that CAQK is specific for brain injuries, as no CAQK accumulation was detected in perforating injuries inflicted on the liver or skin. The reason may be that brain ECM is different from ECM in other tissues in that it mostly consists of a hyaluronan-CSPG-link protein-tenascin complex (Ruoslahti, 1996; Lundell et al, 2004). Thus, CAQK appears to be a specific probe for TBI.

TnC is an extracellular matrix protein that has been shown to affect the proliferation, migration, survival, and differentiation of cells in the oligodendrocyte lineage (Garwood et al, 2004; Kiernan et al, 1996). TnC has a multidomain structure consisting of a coiled-coil region followed by multiple EGF-like domains, fibronectin type-III (FN III) domains, and a fibrinogen C-terminal domain (Okada and Suzuki, 2020). The oligodendrocyte activities have been mapped to the FN (III) repeats in TnC (Freitag et al, 2003). As homing peptides typically bind to ligand-binding sites in proteins (Ruoslahti, 2017; Maldonado et al, 2023), we located the CAQK binding site in TnC to see how it relates to the site that affects oligodendrocytes in TBI. Two TnC fragments—one with only the EGF-like domains and the other with only the FN III (1–6) domains were expressed from cloned cDNAs. The TnC fragment with the FN III (1–6) domains bound FAM-CAQK in the FP assay, whereas the EGF-like domains did not show binding (Fig. 1C). The minimal fragment we found to be positive for dose-dependent FAM-CAQK binding consisted of the FN (III)5 and FN (III)6 domains (Fig. 1D). The TnC binding by CAQK suggested a possible activity in regulating oligodendrocyte responses to injury.

We next tested CAQK targeting and localization in the injured brain of a mouse model of controlled cortical impact (CCI) injury. CAQK showed specific homing to the injury lesion only (Fig. 1E left panel). TnC has limited expression in the adult brain but is rapidly upregulated in inflammatory processes and injuries to the nervous system (Laywell et al, 1992; Okada and Suzuki, 2020). We confirmed high expression of TnC in the injured region of the mouse brain using immunofluorescence analysis. TnC expression in TBI coincided with FAM-CAQK homing after intravenous (i.v.) injection (Fig. 1E right panel). In comparison, the contralateral side of the brain showed minimal TnC expression and no peptide homing. We also observed elevated TnC expression in human injured brain tissue compared to normal brain (Fig. 1F), supporting its potential utility for therapeutic application in humans.

## Pharmacokinetic analysis and brain exposure of CAQK

Given the CAQK accumulation in the injured brain and binding to TnC in injured ECM, and the functional activities observed with other homing peptides, it seemed possible that CAQK might affect the course of TBI. To prepare for functional testing of CAQK, we performed CAQK half-life analyses using a mouse model of CCI. CAQK quantified by mass spectrometry showed biphasic clearance with an initial rapid clearance from the blood (T1/2 = 9 min; Fig. 2A). Initial clearance from the circulation through glomerular filtration is expected for a small peptide, explaining the short half-life. A small percentage of the peptide was eliminated with T1/2 = 4.1 h (Fig. 2B), likely because of covalent binding of CAQK to plasma proteins, such as albumin, through the free sulfhydryl group (Pang et al, 2014) (Fig. EV3). CAQK half-life was also assessed in healthy rats, and it followed a similar biphasic trend as in TBI mice. The half-life remained the same regardless of whether a single injection or daily injections over a week were given (Fig. EV2). At the tissue level, CAQK showed significant accumulation (over eightfold higher) in the injured side of the brain than the contralateral side (Fig. 2C). More than 50% of the initially bound peptide remained in the brain after 2 h with an estimated half-life of about 5.8 h. We have previously shown that FAM-CAQK is still detectable by fluorescence in the injury area after 3 h, confirming the prolonged retention at the target (Mann et al, 2016). This depot effect, likely due to the binding of CAQK to TnC, suggested that a logistically manageable daily injection schedule could be effective for treatment studies.

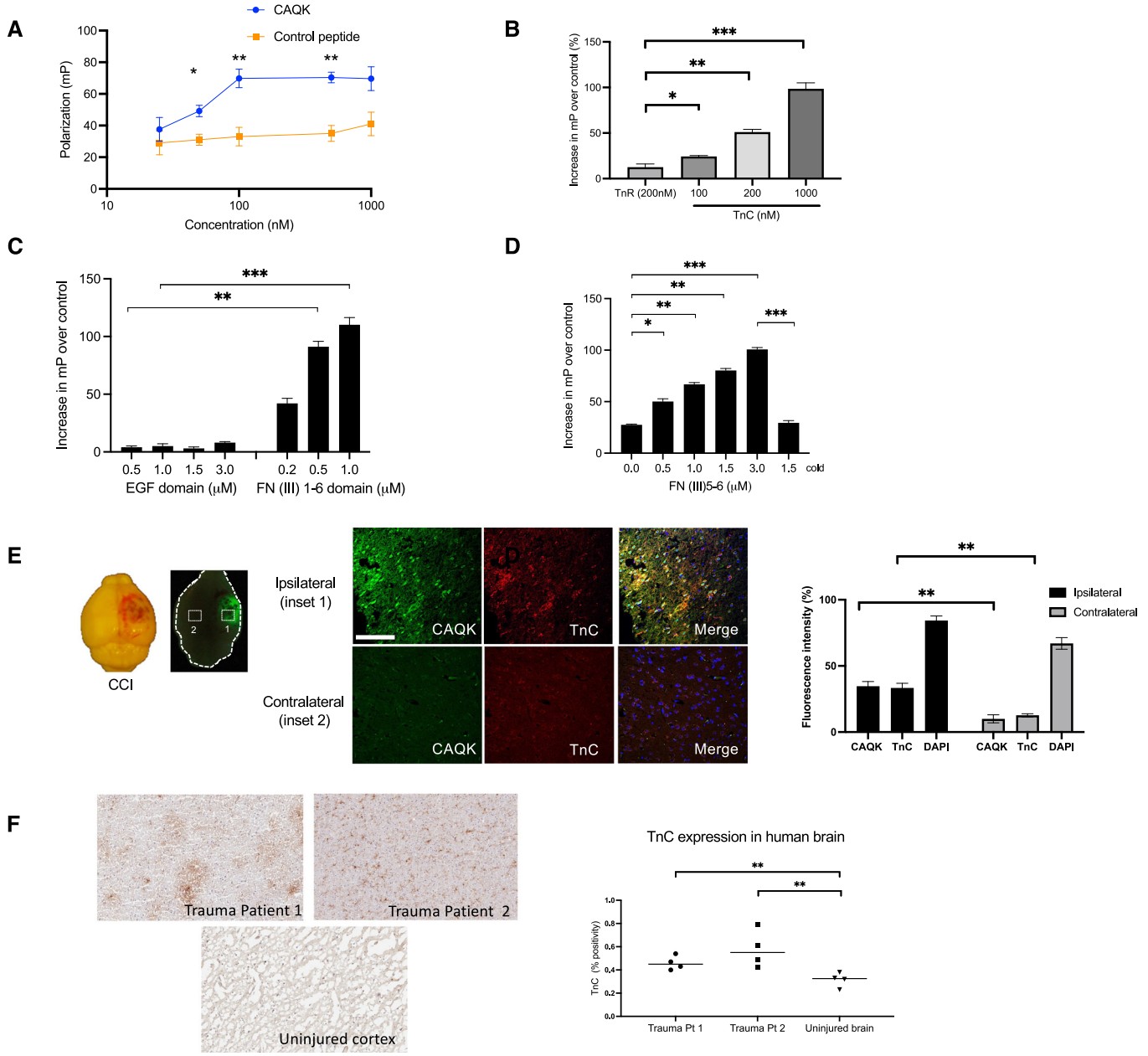

**Figure 1. CAQK binds to TnC in the ECM.**

(**A**) FAM-CAQK binding to purified CSPG complex isolated from chicken brain was analyzed by fluorescence polarization of FAM label on the peptide. FAM-CAQK (20 nM) was incubated with CSPG complex for 60 min at 37 °C. Binding was analyzed by measuring the increase in millipolarization units (mP). FAM-AAQK was used as the control peptide. *$P = 0.02617$, **$P = 0.03684$ (Fisher's exact test). Data expressed as mean ± SEM from three independent experiments. (**B**) Fluorescence polarization of FAM was analyzed to determine FAM-CAQK (20 nM in PBS) binding to purified human TnC after incubation at indicated concentrations for 60 min at 37 °C. Data expressed as mean ± SEM and analyzed using an unpaired *t*-test. *$P = 0.049$, **$P = 0.0071$, ***$P = 0.0037$. $n = 3$. (**C**) CAQK binding to two recombinant TnC fragments was analyzed by fluorescence polarization of FAM on the peptide. FAM-CAQK (20 nM in PBS) was incubated with fragments representing the EGF domains or FN(III) domains of TnC at the indicated concentrations (μM) for 1 h at 37 °C. Data expressed as mean ± SEM and analyzed using an unpaired *t*-test. **$P < 0.01$, ***$P < 0.005$. $n = 3$. (**D**) To localize the CAQK-binding site more accurately, FAM-CAQK (20 nM in PBS) was incubated with a recombinant fragment containing the TnC FN(III) 5–6 domains at indicated concentrations (μM) for 1 h at 37 °C and fluorescence polarization of FAM was analyzed. Cold = excess unlabeled peptide (1 μM) added to TnC fragment. Data expressed as mean ± SEM. Differences were analyzed using an unpaired *t*-test. *$P < 0.05$, **$P < 0.01$, ***$P < 0.005$. $n = 3$. (**E**) Brightfield and fluorescence images of gross mouse brain with CCI and injected with FAM-CAQK at 24 h after injury. Coronal. Left panel shows the intrinsic fluorescence of FAM-CAQK seen macroscopically using an Illumatool bright light system in the green channel. Higher magnification regions (injured and uninjured) are shown stained with anti-TnC antibody (red), FAM (CAQK; green), and nuclei (DAPI, blue). Scale bar, 40 μm. Quantification of positive staining (% area) was performed using Image J. Data were expressed as mean ± SEM. Differences were analyzed using Welch's *t*-test (**$P < 0.01$, **$P = 0.0059$, **$P = 0.0052$). $n = 3$. (**F**) Immunohistochemical staining for TnC expression in cortical brain sections from frozen human brain tissue obtained from two patients with head trauma (Trauma 1 and 2) compared to cortical sections of the brain in a patient without TBI (control). Data expressed as median of five regions of interest. Differences were analyzed using an unpaired *t*-test (**$P = 0.0233$). $n = 1$.

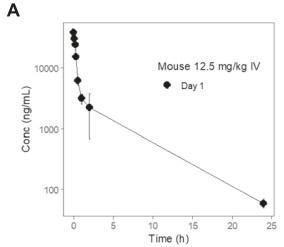

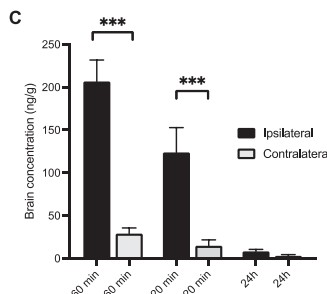

**Figure 2. Pharmacokinetic analysis of CAQK.**

(A) Mice with CCI received i.v. bolus of CAQK (12.5 mg/kg) at 4 h after CCI. Blood was collected at different time points, and plasma was analyzed by LC/MS. Plasma concentration of CAQK was plotted using WinNonlin. $N = 3$ per time point. (B) Plasma clearance data were used to calculate pharmacokinetic parameters using a two-compartment analysis. (C) CAQK accumulation was analyzed in brain homogenates from injured (ipsilateral) and uninjured (contralateral) sides at different time points after i.v. injection and plotted as ng/g of brain tissue. $N = 3$/group. Data were expressed as mean ± SEM. Differences were analyzed using an unpaired $t$-test. ***$P = 0.0216$ vs contralateral side.

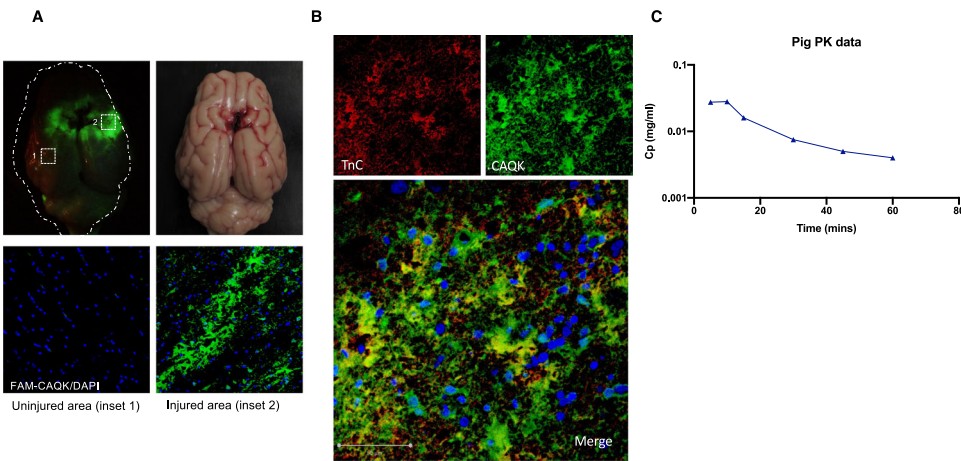

**Figure 3. CAQK targets brain injury in pig CCI.**

(A) FAM-CAQK (2.5 mg/kg) was injected i.v. into pigs with CCI at 1 h post-injury. The peptide was allowed to circulate for 60 min. The pigs were perfused, and brains were isolated and imaged using an Illumatool Bright Light System in the green channel. Intrinsic fluorescence of FAM-CAQK is seen macroscopically. Bottom panel shows immunofluorescence staining of FAM signal (green) in the uninjured area (lower left panel) and injured area (lower right panel). $N = 2$. (B) Fluorescence imaging on cortical brain sections from CCI pig injected with FAM-CAQK. Sections were immunostained with anti-TnC antibody (red), FAM (CAQK; green) and counterstained for nuclei with DAPI (blue). Scale bar, 50 μm. (C) Plasma concentration (Cp) of FAM-CAQK plotted over time. FAM-CAQK (2.5 mg/kg) was injected i.v. into a pig with CCI at 1 h post-injury. Plasma concentration was detected by measuring fluorescence from the FAM label in the peptide at different time points.

## CAQK targeting in a gyrencephalic animal model of TBI

Failure to translate promising TBI therapies in mice, to humans, has been attributed to differences in brain anatomy and physiology between rodents and humans. To determine whether CAQK targets brain injuries in an experimental animal that has a gyrencephalic brain similar to the human brain, we first tested TnC expression in pig brains subjected to CCI. Immunohistochemistry showed robust TnC expression in the brains of CCI pigs not seen in sham-injured pigs (Fig. EV3) and i.v.-injected FAM-CAQK homed to the injured area (Fig. 3A). There was minimal or no homing of CAQK to uninjured parts of the brain (Fig. 3A), or of a control peptide, FAM-AAQK, to the injured brain (Fig. EV3B). CAQK accumulation in the injured brain colocalized with TnC expression in the

injury (Fig. 3B). As in mice, the kidneys were the only normal organ positive for FAM-CAQK (Fig. EV3C). The half-life of FAM-CAQK peptide in pig blood was about 30 min (Fig. 3C). These results indicate that pigs are an appropriate species for CAQK studies.

## Neuroprotective effects of CAQK in TBI

To test CAQK for possible effects on brain injury, mice with CCI were treated with repeated i.v. injections of CAQK (2.5 mg/kg per injection) or vehicle control over a period of 7 days post-injury, the time the blood–brain barrier, (BBB) remains compromised after injury and enables CAQK entry to the lesion (Mann et al, 2016). Macroscopic and histological examination of the brains at day 7 showed greatly reduced

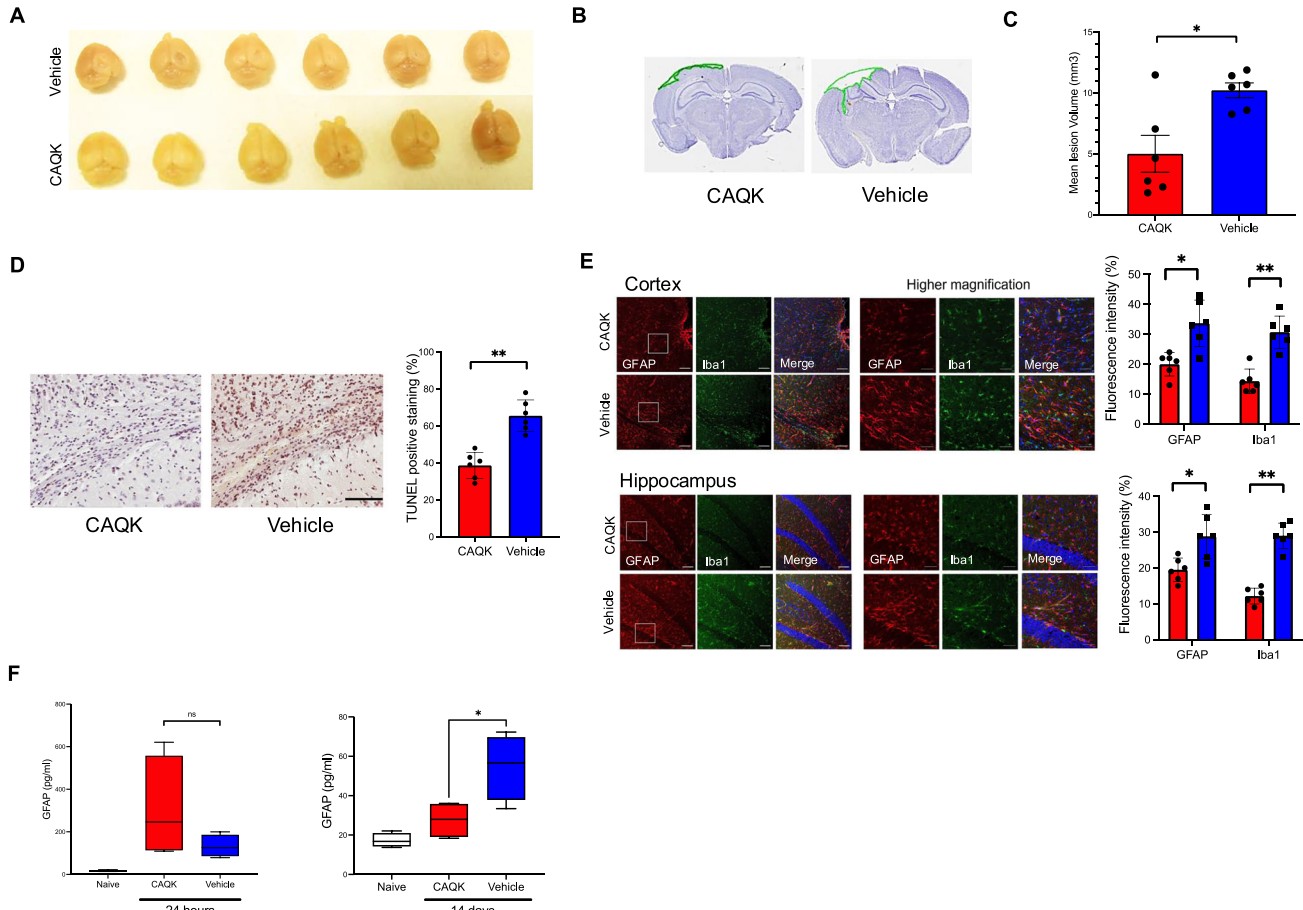

**Figure 4. CAQK effect on TBI in mice.**

(A) Mouse brains were isolated at 7 days post-injury after i.v. treatment with either CAQK peptide (200 nmol (5 mg/kg) per injection) or vehicle (saline) at 6 h after injury and once daily for 7 days. (B) Stereological analysis of coronal brain sections (cresyl violet staining) after treatment with CAQK at post-injury day 7 shows significantly reduced tissue loss (n = 6/group). (C) Quantification of cortical lesion volumes, including the anterior and posterior margins of the contusion, from the stereological analysis in (B). The results indicate a significant difference in lesion areas between the saline-treated and CAQK-treated mice (n = 6/group). Data were expressed as mean ± SEM. Significance was calculated using Fisher's LSD test (*P = 0.0166). (D) Effect of CAQK treatment on apoptotic cell death examined by TUNEL staining (brown) in an area surrounding commissural fibers of the corpus callosum. Scale bar, 100 μm. Quantification of TUNEL staining (% area) was performed using Image J. Data were expressed as mean ± SEM. Differences were analyzed using Fisher's LSD test (**P = 0.001). n = 6/group. (E) Effect of treatment on astrogliosis and microglial activation in injured brain. Coronal sections showing cortex and hippocampus from TBI mice treated with CAQK or vehicle were immunostained for GFAP (red) and Iba1 (green). DAPI staining of nuclei is blue. Left panel shows lower magnification. Scale bar, 100 μm. Insets were magnified and shown on the right panel. Scale bar, 50 μm. Quantification of positive staining (% area) was performed using Image J. Data were expressed as mean ± SEM. Differences were analyzed using Fisher's LSD test (*P = 0.003, **P = 0.001). n = 6/group (F) Serum levels of GFAP in TBI mice at 24 h and 14 days after brain injury are plotted in a box and whiskers plot with whiskers down to the minimum, and up to the maximum, values. Data compared to naïve mice with no injury and no treatment. Plasma concentration shown (pg/ml; n = 5/group). Data were expressed as mean ± SEM. Differences were analyzed using an unpaired t-test (*P = 0.0294).

(about 50%) tissue loss compared to controls (Fig. 4A–C). CAQK-treated mice also showed significantly reduced apoptosis, as measured by TUNEL staining in the injured areas (Fig. 4D). Neuroinflammation is a major element of the secondary response after TBI. It is associated with reactive gliosis, which is characterized by glial hypertrophy, increased expression of the glial-specific intermediate filament protein GFAP (glial fibrillary acidic protein), and microglial activation (Cederberg and Siesjo, 2010; Ramlackhansingh et al, 2011). We observed a drastic reduction in GFAP expression, which reflects the number of activated astrocytes in the lesions and peri-lesion area of CAQK-treated mice at day 7 post-injury (Fig. 4E). There was also a significant reduction in the microglia activation marker, Iba1 (Fig. 4E).

GFAP is also a commonly used neurological blood biomarker that has been studied as an indicator of TBI severity (Korley et al, 2018). We tested whether GFAP might be a suitable surrogate marker of treatment efficacy. As expected, all mice with TBI, regardless of treatment, showed a dramatic increase in plasma GFAP at 24 h after injury in all TBI mice. However, at 14 days after injury, the CAQK-treated mice had GFAP levels significantly lower than the vehicle-treated mice (Fig. 4F). Another plasma biomarker, NF-L that has been associated with TBI severity, was also tested at 24 h and 14 days post-TBI. Although NF-L levels at day 14 in the CAQK-treated group trended lower than vehicle group, the difference was not statistically significant (Appendix Fig. 1). Thus, plasma GFAP levels at post-acute phase (day 14) support the

accelerated healing of CAQK indicated by the neuroinflammatory markers examined above.

To investigate further the apparent attenuation of neuropathology and neuroinflammation after CAQK treatment, we performed a gene profiling analysis of the TBI lesions. As expected, gene expression associated with neuroinflammation, and neuropathology was robustly increased in TBI mice compared to naïve mice. Volcano plots show the number of neuroinflammation-associated genes that were differentially expressed in vehicle-treated and naïve mice (Fig. 5A, left panel) and CAQK-treated and naïve mice (Fig. 5A, right panel). Similar pattern on changes in gene expression was observed for the neuropathology panel (Appendix Fig. S2). CAQK treatment reduced the number of differentially expressed genes by half relative to the vehicle-treated group (Fig. 5B). A heat map of the highest fold change and statistically significant differential gene expression between CAQK treated vs vehicle-treated mouse brains is shown in Fig. 5C. Pathway analysis showed that the most prominent pathways affected by CAQK were the TYROBP causal network in microglia, and microglial phagocytosis, complement activation and inflammatory response pathways (Fig. 5D). All these pathways are highly correlated with TBI severity and even serve as prognostic indicators (Rana et al, 2019). Key inflammatory genes known to promote inflammation by enhancing M1 microglial polarization—including Lipocalin 2, TGFβ, complement C3, CXCL10, TNF, and S100b—showed significant downregulation after CAQK treatment (Fig. 5E). Interestingly, TnC, the target for CAQK, was also significantly reduced, reaching close to naïve levels following CAQK treatment. These results show that CAQK treatment reverses TBI-related expression of genes associated with inflammation, interferon signaling and neuropathology, reinforcing the other data in building a case for CAQK-mediated improvement of recovery from TBI.

The main goal of any treatment of TBI is the best possible functional recovery. An initial "neuroscreen" of TBI mice performed at day 10 showed that the mice had difficulty holding onto the cage top, suggesting the existence of motor problems that would allow comparison of the CAQK-treated and control-treated mice. Reflexes tested by evaluating ear twitch, whisker, and toe pinch responses also suggested signs of recovery of the CAQK-treated mice compared to vehicle-treated mice (Appendix Table 1).To assess motor coordination, rotarod and hanging wire tests were conducted on days 15–18 and days 22–23, respectively (Crawley, 2007; Freitag et al, 2003). In both tests, the CAQK group showed significant recovery compared to vehicle-treated controls (Fig. 6A,B). In a novel object recognition test performed on days 25–29 after TBI, CAQK-treated mice showed greater preference for the novel object than the vehicle-treated group (Fig. 6C), also indicating improved recovery of cognitive abilities.

Finally, to assess any potential safety issues, we conducted preliminary safety studies in TBI mice and in healthy rats. The animals were treated daily with different i.v. doses of CAQK up to 300 mg/kg. There were no effects on body weight, organ weight, or clinical pathology (Appendix Tables 2 and 3).

In aggregate, these results show that CAQK treatment results in significant improvement in TBI lesion size and neurological function in mice, suggesting the possibility that CAQK could offer a new treatment option for the management of brain injuries.

## Discussion

We describe an inherent neuroprotective role of CAQK, a tetrapeptide that was selected to accumulate in areas of brain injury upon intravenous administration and envisioned as a carrier to be used in site-specific delivery of drugs to brain injuries (Mann et al, 2016). We show that tenascin-C, an ECM component which is strongly upregulated in brain injury, is the molecule that binds CAQK in TBI lesions. We also describe an inherent neuroprotective role for CAQK. CAQK alone, with no drug attached to it, improves the recovery of mice from acute TBI by histological, biomarker and behavioral criteria. This activity is likely to be a result of the binding of CAQK to TnC. The neuroprotective activity and the demonstration that CAQK also recognizes injuries in gyrencephalic brains in humans and pigs, suggests that this simple and well-defined compound has potential as a new treatment for brain injuries and warrants further study.

Several lines of evidence support the identification of TnC as the target molecule. First, TnC was the only CAQK-binding molecule present in substantial quantities in the two sources we used to prepare ECM. Second, further analysis showed direct binding of FAM-CAQK to recombinant human TnC and to fragments that encompass FN (III) domains 5–6 of TnC. Third, TnC is strongly overexpressed in the ECM of injured regions of the mouse and human brain (Mann et al, 2016; Shiba et al, 2014), and CAQK accumulation in brain injuries colocalized with TnC.

TnC, the protein we identify as the CAQK target molecule in brain injury ECM is a regulator of multiple cellular functions (Shiba et al, 2014) suggesting that the TnC binding is likely to be functionally significant beyond making possible the accumulation of CACK at sites of injury. TnC contributes to glial scarring and causes neuroinflammation, blood–brain barrier disruption, and neuronal apoptosis (Okada and Suzuki, 2020). TnC also inhibits oligodendrocyte differentiation and myelin basic protein (MBP) expression during recovery from a demyelination injury (Czopka et al, 2009),(Czopka et al, 2010) (Bauch and Faissner, 2022). The underlying signaling cascade requires the binding of the cell surface receptor contactin1 to TnC FN(III) 5 and 6 domains (Weber et al, 1996). Our data on CAQK binding to TnC FN(III) 5–6 domains suggests that CAQK could impact these functions of TnC. Further studies are needed to evaluate CAQK activity on oligodendrocyte maturation and differentiation to confirm this effect.

Our study shows that CAQK therapy administered intravenously within hours after TBI in mice reduced the lesion size, evident both in inspection of the brain and microscopic analysis. The reduction in apoptosis we observed is a likely explanation for the reduced lesion volume. Neuroinflammation was also reduced, judging from lower levels of inflammatory markers, which included the microglial marker Iba1 and pathways associated with microglial activation. The strong reduction of Iba1 is particularly noteworthy because human and animal studies have shown that microglia can be chronically activated for weeks to years after brain trauma (Gentleman et al, 2004; Johnson et al, 2013; Nagamoto-Combs et al, 2007; Nonaka et al, 1999).

TBI initiates a complex cascade of dysregulated pathways that requires a multifunctional therapy. CAQK affected so many different events in the tissue response to TBI that it seems capable of providing the type of pluripotent effect needed to deal with multiple pathways. It may be that CAQK modulates a pathway that

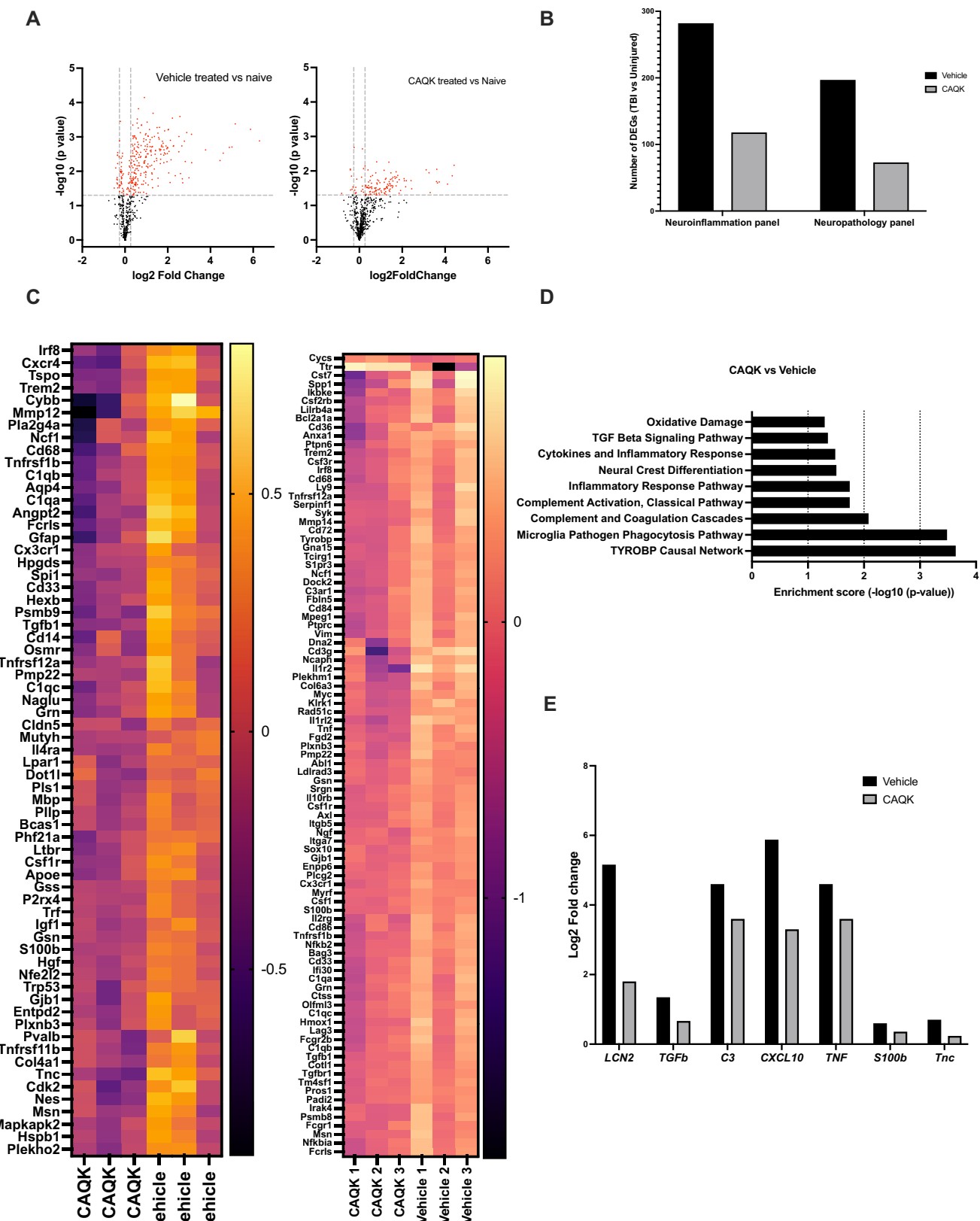

**Figure 5. CAQK treatment reduces expression of neuroinflammatory genes in TBI.**

(A) Volcano plots showing −log10(*p* value) and log 2-fold change in genes that were expressed differentially at a significant level (fold increase >1.2 or fold decrease <1.2 and *P* < 0.05) in vehicle-treated and naïve mice (left panel) and CAQK-treated and naïve mice (right panel) using the neuroinflammation panel (*n* = 3). The generalized linear model (GLM) developed by NanoString is used for calculating differential expression for count data. (B) Number of differentially expressed genes (DEGs) in injured brains at day 14 after injury compared to 12-week-old naïve brains from uninjured mice. Genes were filtered based on fold change >1.20, *P* < 0.05 as a threshold. (C) Nanostring-based heatmaps showing differential gene patterns in CAQK-treated and vehicle-treated groups using the neuroinflammation panel (left panel) and neuropathology panels (right panel). Color coding was based on z-score scaling. (D) Pathway analysis of the genes that differed most in the neuroinflammation panel between CAQK-treated and vehicle-treated mice after CCI. Differentially expressed genes based on fold change >1.20, *P* < 0.05 as threshold. Gene set analysis (GSA) module from NanoString® used for statistical analysis. (E) Analysis of key inflammatory genes implicated in TBI arranged in order of fold change in the vehicle-treated group. LCN2 (lipocalin 2), Tgfb1 (transforming growth factor-beta 1), C3 (complement factor 3), CXCL10 (C-X-C Motif Chemokine Ligand 10), TNF (tumor necrosis factor), S100b (S100 Calcium Binding Protein B), and TnC (tenascin-C).

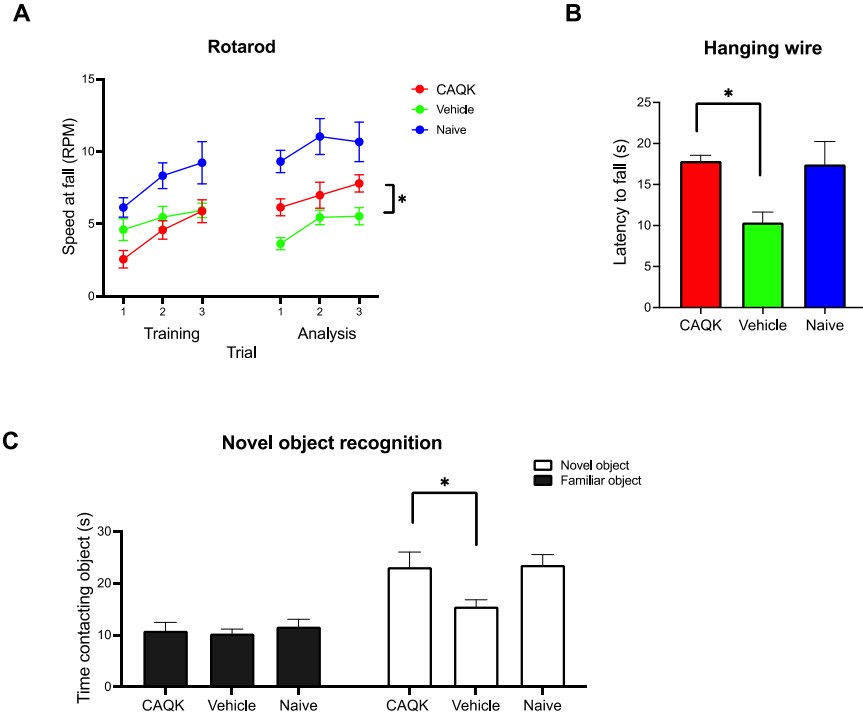

**Figure 6. CAQK treatment improves behavioral outcome in TBI mice.**

(A) Rotarod test performed on TBI and naive mice at day 15–18 after injury. Results expressed as mean rotational velocity in rpm at the time of falling off the rod in three successive tests. *n* = 10 per group. **P* = 0.0246, CAQK treated vs. vehicle control (ANOVA). (B) Hanging wire test performed in mice on days 22–23 after injury. The latency from the beginning of the test until the time the mouse fell off was timed up to a maximum of 30 s, and each animal got three trials separated by 30 s. *n* = 8 per group. Data were expressed as mean ± SEM. **P* = 0.0393. Fisher's LSD test for CAQK-treated vs. vehicle control. (C) Novel object recognition test in mice on days 25–29 after injury. *n* = 10 per group. Data were expressed as mean ± SEM. **P* = 0.0246, one-way ANOVA with Bonferroni test.

is central to the initial response to TBI, and that this hypothetical pathway is governed by TnC.

The results presented here are limited to intravenous delivery of CAQK. Short peptides such as CAQK are also amenable to other routes of administration, such as intranasal delivery. Future studies to explore alternative routes of delivery and different dosing intervals can extend the utility of CAQK for TBI management. For prolonged brain residence, further studies can also evaluate the D-enantiomer version of CAQK to improve stability against enzymatic degradation, which could enhance the therapeutic duration and reduce the frequency of administration of CAQK.

Importantly, CAQK therapy resulted in improvement in the functional recovery of TBI mice. The behavioral effects of CAQK

were significant but not as striking as the effects on lesion size and tissue markers. Rodents have relatively small lissencephalic brains in which white matter is less than 20% compared to 80% in large animals and humans; therefore, only limited white matter axonal pathology is observed in rodent TBI models (Ventura-Antunes et al, 2013). This circumstance has made it difficult to translate mouse results to the clinic. The results reported here show that CAQK also recognizes brain injuries in pigs, a species with a gyrencephalic brain similar to the human brain. We also know from our earlier study (Mann et al, 2016) that CAQK binds to injured, but not normal, human brain in microscopic overlay assays. Thus, the main prerequisites for conducting efficacy studies in pigs that can include clinically relevant imaging readouts, like

MRI, to better assess edema or vascular conditions and any subsequent clinical trials are in place.

An important consideration regarding the use of CAQK in treating brain injuries is access to extravascular brain tissue, where CAQK binds to the ECM and is retained for a prolonged period of time. The ECM binding may provide a reservoir and favorable localization for CAQK that can potentially amplify its functional effects. Moderate to severe TBI disrupts the structural and physiological integrity of blood vessels resulting in an impaired blood–brain barrier (BBB) that starts as early as 30 min after injury, and that can last up to a week after the injury (Barzo et al, 1996; Mann et al, 2016), providing a window for systemically administered CAQK to enter the injured target tissue. Disruption of BBB has been observed in TBI patients in the acute phase and given that human brain is not as resilient as mouse brain, the BBB dysfunction may persist well into the chronic post-injury phase (Hay et al, 2015) (Stahel et al, 2001). This can provide a much longer treatment window for CAQK than is available in mice. In addition to TBI (Li et al, 2022; Wu et al, 2019), specific homing of CAQK has been reported in other central nervous system lesions, including spinal cord injury (Wang et al, 2022; Wang et al, 2020; Wang et al, 2018b), and demyelinating injuries (Abi-Ghanem et al, 2022). If the BBB is compromised in these conditions, as it is in TBI, they could potentially also be treated with CAQK.

The tools available for clinicians to treat TBI patients are limited. The current clinical practice in treating TBI patients is essentially limited to supportive care, including the vitally important decompressive craniectomy or other decompression protocols in severe cases of TBI. Cerebrolysin, a peptide preparation generated by controlled digestion of porcine brain proteins, is approved for human use in several countries, but has shown only modest improvement of recovery as measured using the Glasgow coma scale (Lucena and Briones, 2022). Another deterrent to its use may be the ill-defined nature of the compound. CAQK, in contrast, is a simple, well-defined compound. N-acetyl-L-cysteine (NAC) (Hegdekar et al, 2021; Koh et al, 2018; Sakane and Pardridge, 1997), has some similarity with CAQK, as both are small compounds with a free thiol group. The thiol group may contribute to scavenging of tissue-damaging oxygen radicals at the site of injury (Khatri et al, 2018; Di Pietro et al, 2020). The levels of the physiological antioxidant, glutathione (GSH) and its precursors decrease following TBI (Koza and Linseman, 2019). The thiol group of GSH is supplied by cysteine. Thus, NAC used to treat liver toxicity caused by acetaminophen (Heard, 2008) could either act as a direct ROS scavenger or increase the supply of cysteine for the replenishment of GSH (Eakin et al, 2014). CAQK, with its free thiol-containing cysteine, may act in a similar fashion.

Given the heterogeneity in presentation and severity of TBI patients, utilization of blood-based protein biomarkers such as GFAP has been approved and validated for initial patient evaluation. GFAP levels at early time points correlate with injury severity and are of prognostic value in functional outcome in TBI cases (Wang et al, 2018a). Our results in TBI mice showed that GFAP levels peaked at 24 h in both vehicle and CAQK-treated groups, indicating comparable severity of TBI. It is unlikely to expect significant improvement of GFAP levels within 24 h and only one CAQK injection. However, after 14 days, reduced GFAP levels post injury in the CAQK-treated group compared to the vehicle group suggest that GFAP could be used to monitor

treatment efficacy in future clinical studies. Similarly, multiple studies have shown that serum tenascin-C concentrations of TBI patients are significantly elevated and negatively correlated with favorable outcomes (Minta et al, 2019; Zhao et al, 2017). Our rodent studies show that CAQK treatment results in a reduction in TnC expression in the injured brain, suggesting that TnC levels might be a potentially useful pharmacodynamic biomarker.

In conclusion, treatment of TBI mice with CAQK peptide strikingly limits secondary tissue damage from the injury. This, and its apparent lack of toxicity, are encouraging and suggest that further studies toward clinical translation of CAQK are warranted.

# Methods

**Reagents and tools table**

| Reagent/resource | Reference or source | Identifier or catalog number |
|---|---|---|
| **Experimental models** | | |
| U251 | ATCC | 9063001 |
| **Antibodies** | | |
| Rat anti-tenascin-C | R&D systems | MAB2138 |
| Rabbit anti-fluorescein/ Oregon Green | Thermo Fisher | A889 |
| Rat anti-GFAP | Thermo Fisher | 13-0300 |
| Rabbit anti-Iba1 | FujiFilm | 019-19741 |
| Mouse IgG control | Thermo Fisher | 02-6502 |
| **Chemicals, enzymes, and other reagents** | | |
| Fetal bovine serum (FBS) | Thermo Fisher | 10270106 |
| DPBS | ThermoFisher | 14190169 |
| high glucose DMEM | Thermo Fisher | 41965062 |
| n-octyl-beta-D-glucopyranoside | MilliporeSigma | 494459 |
| cOmplete™, EDTA-free Protease Inhibitor Cocktail | MilliporeSigma | 04693132001 |
| SulfoLink™ Coupling Resin | Thermo Fisher | 20401 |
| Phenol-red free DMEM | Thermo Fisher | 21063029 |
| HiTrap DEAE Sepharose FF | Cytiva | 17505501 |
| Chicken chondroitin sulfate proteoglycans (CSPG) mixture | MilliporeSigma | CC117 |
| Pierce™ BCA Protein assay | Thermo Fisher | 23225 |
| Trizol | Thermo Fisher | 15,596–018 |
| HRP-DAB TUNEL assay kit | Abcam | ab206386 |
| Chondroitinase ABC enzyme | AMS Bio | AMS.E1028-02 |
| **Software** | | |
| Origin 2022b | https://www.originlab.com/2022b | |

| Reagent/resource | Reference or source | Identifier or catalog number |
|---|---|---|
| WinNonlin | https://www.certara.com/software/phoenix-winnonlin/ | |
| Image J software | https://imagej.nih.gov/ij/index.html | |
| nSolver software Version 4.0 | https://nanostring.com/products/ncounter-analysis-system/ncounter-analysis-solutions/ | |
| Imagescope software | Leica Biosystems | |
| Rosalind software | https://www.rosalind.bio | |
| Statistica 8.0 software | https://statistica.software.informer.com/8.0/ | |
| GraphPad Prism 10 | https://www.graphpad.com | |
| Other | | |
| PHERAstar FS | BMG Labtech | |
| Illumatool Bright Light System LT-9900 | Lightools Research | |
| Zeiss LSM-710 confocal microscope | Zeiss Group | |

## Controlled cortical impact model

All experiments for the CCI model were conducted under an approved protocol (AUF#19-077) of the Institutional Animal Care and Use Committee of Sanford Burnham Prebys Medical Discovery Institute and all experiments were performed in accordance with relevant guidelines and regulations. The ARRIVE guidelines for reporting were followed. The CCI model was used as described (Krajewska et al, 2011). Eight- to ten-week-old male C57BL/6 mice were anesthetized with 4% isoflurane (Aerrane; Baxter, UK) in 70% $N_2O$ and 30% $O_2$ and positioned in a stereotaxic frame. Using a head restraint, a 5-mm craniotomy was made using a portable drill and a trephine over the right parietotemporal cortex and the bone flap was removed (Kielian et al, 2001). Mice were subjected to CCI using the benchmark stereotaxic impactor (Impact One™; myNeuroLab.com) with the actuator part mounted directly on the stereotaxic instrument. The impactor 3 mm tip accelerated down to the 1.0 mm distance, reaching the preset velocity of 3 m/s, and the applied electromagnetic force remained there for the dwell time of 85 ms, and then retracted automatically. The contact sensor indicated the exact point of contact for reproducible results. Face mask anesthesia (1–2% isoflurane in 70/30% nitrous oxide/oxygen) was used during the entire procedure. In both cases, afterwards, the scalp was closed with sutures, anesthesia was discontinued, and mice were administered buprenorphine i.p. for pain control. For the first 2 h post-CCI, mice were closely monitored in their cages.

## Peptide synthesis

The peptides were synthesized on a microwave-assisted automated peptide synthesizer (Liberty; CEM, Matthews, NC) following Fmoc/t-Bu (Fmoc:Fluorenyl methoxy carbonyl, t-Bu: tertiary-butyl) strategy on Rink amide resin with HBTU (N,N,N′,N

′-Tetramethyl-O-(1H-benzotriazol-1-yl)uranium hexafluorophosphate (OR) O-(Benzotriazol-1-yl)-N,N,N′,N′-tetramethyluronium hexafluorophosphate)activator, collidine activator base and 5% piperazine for deprotection. Fluorescein tag was incorporated during synthesis at the N-terminus of the sequence for homing studies. Cleavage using a 95% TFA Trifluoroacetic acid followed by purification gave peptides with >90% purity. Peptides were lyophilized and stored at −20 °C.

## Affinity chromatography and proteomics

For identifying CAQK binding proteins, mouse brains with CCI were collected 6 h post-injury. Using liquid nitrogen, the brains were crushed and ground into powder in a mortar and pestle. Next, brain tissue was lysed in PBS containing 200 mM n-octyl-beta-D-glucopyranoside and protease inhibitor cocktail (Roche) as described previously (Teesalu et al, 2009) with slight modifications. The clarified lysates were loaded onto CAQK-coated Sulfolink-agarose beads (Pierce) and incubated at 4 °C for 3–4 h. The column was washed with wash buffer, followed by washing with 0.5 mM CGGK control peptide to remove nonspecifically bound proteins. The bound proteins were eluted with 2 mM free CAQK peptide. The eluted fractions were pooled, their protein concentration determined by using bicinchoninic acid (BCA) protein assay (Pierce) and the samples were digested using the filter-aided sample preparation (FASP) method (Wisniewski et al, 2009). Finally, the digested samples were dried, desalted and subjected to LC-MS/MS analysis at the Sanford Burnham Prebys Medical Discovery Institute's Proteomics Core facility. All mass spectra were analyzed with MaxQuant software version 1.5.0.25. The MS/MS spectra were searched against the Mus musculus Uniprot protein sequence database (version July 2014).

## Proteomics analysis to identify the CAQK binding receptor

Serum-free conditioned medium enriched in proteoglycans was obtained by growing U251 cells in phenol-red-free DMEM culture media. The collected media was centrifuged to remove cell debris. The supernatant was run on an anion exchange diethyl-aminoethyl groups (DEAE) column to capture the proteoglycans. The column was washed with 50 mM Tris-Cl, pH 8.0 and 250 mM NaCl. The purified protein was eluted with two different elution buffers. The elution buffer 1 consisted—8 M Urea, 50 mM Tris-Cl, 150 mM NaCl, and elution buffer 2 consisted—50 mM Tris-Cl, 1 M NaCl. The eluted fractions (0.5 mL) were dialyzed in buffer containing 50 mM Tris-Cl, 100 mM NaCl, pH 7.4. The eluted and dialyzed fractions from the DEAE column were subjected to LC-MS/MS analysis at the Proteomics Core facility at Sanford Burnham Prebys Medical Discovery Institute. Chicken chondroitin sulfate proteoglycans (CSPG) mixture (MilliporeSigma cat #CC117) was digested with Chondroitinase ABC enzyme (AMS Bio, Cambridge, MA) and subjected to LC-MS/MS analysis to identify the binding partner of CAQK peptide.

## Fluorescence polarization assay

Fluorescence polarization (FP)–based assay was used for solution-based binding of CAQK to different proteins. FP assay was initially

set up in 50 µL wells with final volumes in 96-well plates (Corning Life Sciences), and measurements were carried out using PheraStar FS plate reader (BMG LABTECH, Ortenberg, Germany) in triplicate. FAM-Peptide and protein stock solutions were diluted to the desired concentration with assay buffer (10 mM HEPES, pH 7.4, 1 mM $CaCl_2$, 150 mM NaCl, 0.1% Pluronic). FAM-labeled peptides were protected from light during the experiment to avoid photobleaching. In the binding assay, the concentration of FAM-peptide was kept constant, and the FP was measured over the range of target protein concentrations. The specificity of the binding assay was confirmed by the displacement assay, in which FAM-labeled peptide was competed off from the target protein by unlabeled peptide.

## Cell culture and treatment

U251 cells (from ATCC, Manassas, VA, USA) were cultured at 37 °C in 5% $CO_2$ in high glucose DMEM and 10% fetal bovine serum supplemented with penicillin (100 U/ml) and streptomycin (100 µg/ml). Conditioned media were obtained by conditioning the growth media for 24 h with confluent cells immediately prior to passaging. CM were centrifuged at $1500 \times g$ for 10 min and mixed with fresh growth medium at a 1:1 ratio prior to use.

## Homing studies and immunofluorescence

Animals were intravenously (i.v.) injected, 6 h after brain injury, with 50 nmoles of FAM-labeled peptide dissolved in PBS and allowed to circulate for 30 min. Mice were perfused intracardially with saline, and brains were isolated and imaged using the Illumatool Bright Light System LT-9900 (Lightools Research). For fixing the biological material, brains and organs were fixed in paraformaldehyde (4%) at pH 7.4 for overnight and dehydrated in sucrose gradient (15–30%) followed by cryosectioning. Sections were permeabilized using PBS-Triton, and blocking was carried out using 5% blocking buffer: 5% BSA, 1% goat serum (Jackson Immuno Research), 1% donkey serum (Jackson Immuno Research) in PBS-T. Primary antibodies were incubated in diluted (1%) blocking buffer overnight at dilutions 1/100 or 1/200 at 4 °C, washed with PBS-T and incubated with secondary antibodies diluted 1/200 or 1/500 in 1% diluted buffer for one hour at room temperature, subsequently washed with PBS-T, counterstained with DAPI 1 ug/mL in PBS for five minutes, washed with PBS, and mounted using mounting media, and imaged the same day or the day after using Zeiss LSM-710 confocal microscope. Antibodies used: Human anti-tenascin-C (R&D Systems), rat anti-tenascin-C (R&D Systems), Rabbit anti-fluorescein/Oregon Green (Thermo Fisher), GFAP antibody 2.2B10 (Thermo Fisher). Sections were imaged using a Zeiss LSM-710 confocal microscope. Quantification of immunoreactivity was done in a blinded manner by calculating fluorescent intensity in six randomly selected fields of view and processed using Image J software (v1.8.0, NIH, USA).

## Pharmacokinetic analysis

CAQK (12.5 mg/kg) was injected i.v. into mice with CCI at 4 h post-injury. Blood was collected from these mice at different time points, and plasma was processed and analyzed by LC/MS after a reduction and alkylation step under reducing conditions to capture total peptide (free peptide and peptide bound to plasma proteins by disulfide formation). Plasma concentration was plotted using WinNonlin. Similarly, for analysis in rats, CAQK (10 mg/kg) was injected i.v. into healthy Sprague-Dawley rats once daily for 7 days. Blood was collected at different time points at Day 1 and Day, 7 and plasma was analyzed by LC/MS to analyze CAQK. Plasma concentration of CAQK was plotted. $N = 3$ per time point.

## Peptide treatment studies

For treatment studies, mice with CCI were randomized into two groups ($n = 6$/group) and injected with CAQK (200 nmoles dissolved in 100 µl PBS) or vehicle (PBS) alone starting at 6 h after injury. The mice received the same treatment daily until day 7 after injury. At day 10, the mice brains were extracted and placed in 4% paraformaldehyde (PFA) at pH 7.4 overnight, washed with PBS and placed in graded sucrose solutions overnight before optimal cutting temperature compound (OCT) embedding. Ten-micron-thick sections covering the entire injury were cut and analyzed by immunofluorescence. To measure the lesion volume, the sections were stained with Nissl stain, and the lesion area for each distance from bregma was measured using Imagescope software (Leica Biosystems). The lesion area was plotted against the bregma distance and integrated over the entire injury length to obtain the lesion volume using a polynomial fit in Origin 2022b.

## Behavioral testing

Mice with TBI were treated and compared with naïve mice using the following three behavioral tests. Novel object recognition test assayed recognition memory while leaving the spatial location of the objects intact (Winters et al, 2004). Mice were individually habituated to a 51 cm × 51 cm × 39 cm open field for 5 min. Mice were tested with two identical objects placed in the field (two 250 ml amber bottles). Each individual animal was allowed to explore for 5 min, now with the objects present. After three such trials (each separated by 1 min in a holding cage), the mouse was tested in which a novel object replaced one of the familiar objects. Behavior was video recorded and then scored for contacts (touching with nose or nose pointing at object and within 0.5 cm of object). Recognition indexes were calculated using the following formula: # contacts during test/ (# contacts in last familiarization trial + # contacts during test). Rotarod balancing requires a variety of proprioceptive, vestibular, and fine-tuned motor abilities as well as motor learning capabilities (Carter et al, 2001). A Roto-rod Series 8 apparatus (IITC Life Sciences, Woodland Hills, CA) was used, which records test results when the animal drops onto the individual sensing platforms below the rotating rod. An accelerating test strategy was used, whereby the rod started at 0 rpm and then accelerated at 10 rpm. The mice were trained in three sets of three consecutive trials. Five days later, the mice were retested in three consecutive trials. The hanging wire test allowed for the assessment of grip strength and motor coordination (Crawley, 2007; Freitag et al, 2003). Mice were held so that only their forelimbs contacted an elevated metal bar (2 mm diameter, 45 cm long, 37 cm above the floor) held parallel to the table by a large ring stand and were let go to hang. Each mouse was scored in three trials separated by 30 s. Latency to falling off was also measured up to a maximum of 30 s.

## Swine injury studies

This study was approved by the Institutional Animal Care and Use Committee at David Grant USAF Medical Center, Travis Air Force Base, CA (Protocol Number - FDG20210026A). All animal care and use were in compliance with the Guide for the Care and Use of Laboratory Animals in a facility accredited by AAALAC. Castrated male Yorkshire cross swine were acclimated in the facility for at least 10 days before use. They were fed a standard diet and observed for any health problems. Food was withheld the night before surgery, but the pigs had access to drinking water. They were premedicated, intubated, and anesthetized by a trained veterinarian. Premedication and anesthesia induction was achieved with Tiletamine/Zolazepam (Telazol) 6.6–8.8 mg/kg by intramuscular injection, and the patient was endotracheally intubated and mechanically ventilated. Animals were positioned supine, followed by placement of a cardiac monitor, temperature probe, end tidal carbon dioxide monitor, and oxygen saturation monitor. Ventilator settings were adjusted to maintain end tidal carbon dioxide at $40 \pm 5$ mmHg. Ophthalmic ointment was applied to each eye to prevent corneal drying. All animals were kept on a warming blanket set to 39 °C to prevent hypothermia. Additional warm air blowers were used if needed to maintain body temperature at or above 37 °C. Crystalloid fluids were administered at 5 mL/kg/h throughout the experiment. Venous blood was drawn for a preoperative complete blood count (CBC) and complete metabolic profile (CMP). Animals were excluded from further use if their white blood cell count exceeded $25 \times 10^9$/L, they were anemic (PCV<25%), or if their blood chemistry (blood urea nitrogen, creatinine, alanine transaminase, aspartate transaminase, or total protein) values were elevated beyond reference intervals. The craniotomy and vascular access sites were clipped and cleaned of any gross debris. A femoral artery was cannulated percutaneously for the placement of an arterial line for blood pressure measurements and blood gases. A vascular line was placed in the left external jugular vein for venous blood collection and fluid/drug administration. In the prone position, the skin overlying the frontal region of the skull was incised, and the soft tissues dissected to expose the skull. A 21 mm burr hole was created on the right frontal bone, 3 mm lateral to the sagittal suture, and 7 mm rostral to the coronal suture, taking care not to lacerate the dura. A 2 mm burr hole was made 10 mm lateral and 10 mm caudal to the bregma for insertion of a brain intracranial pressure monitor. The stereotaxic frame was positioned over the head and secured with stabilizer rods inserted into the skull. The exposed cortex was injured with a controlled cortical impact device (Custom Design and Fabrication, Inc., Richmond, VA) mounted to the stereotaxic frame. The parameters include a 20 mm rounded polymer striking tip, 4 m/sec velocity, 12 mm depth, and 200 µs dwell time. The craniotomy was packed with bone wax, and the skin was sutured in place with continuous sutures. Serial blood samples were drawn from an artery to perform baseline arterial blood gas analyses, CBC, CMP, viscoelastic testing, and to obtain sera for later analyses at 5, 10, 15, 30 min, and then hourly for 6 h. Standardized supplemental fluids, norepinephrine infusions, and correction of any electrolyte and glucose abnormalities was performed. A minimum MAP of 65 mmHg was maintained using 500 mL crystalloid boluses and norepinephrine infusions (8 mg/250 mL normal saline) starting at 0.01 mcg/kg/min, up to 0.2 mcg/kg/min. Goal electrolyte and lactate concentrations were monitored and corrected as needed. One hour after injury 2 animals were intravenously injected with fluorescently labeled peptide (2.5 mg/kg, equivalent to a mouse dose of 5 mg/kg), delivered as a bolus over 10 min by intravenous infusion pump. Animals were monitored for 1 h following the completion of treatment, during which time serial blood samples were drawn as described above. Animals were humanely euthanized by barbiturate overdose 1 h after the completion of treatment/placebo administration. Perfusion with 3 L phosphate-buffered saline was performed by cannulating both carotid arteries. The brains were extracted, chilled at −80 °C for 30 min, and sliced. Tissue from the heart, lung, liver, intestine, kidney, spinal cord, and adductor muscle was collected and fixed in formalin. The brain was sliced through the injury impact area in 5 mm coronal slices using a brain block (Zivic Instruments, Inc., Pittsburgh, PA). Brain slices were immersion-fixed in 4% (w/v) paraformaldehyde/PBS for 24 h. Brain slices were transferred to fresh 4% paraformaldehyde for 48 to 72 h at 4 °C with gentle shaking and then transferred to PBS for 24 h at 4 °C, followed by dehydration in sucrose gradient (15–30%) followed by cryosectioning at 10–20 µm for immunohistochemistry.

## Safety analysis

Male C57BL/6 mice ($n = 5$ per group) with CCI injury were intravenously administered daily with control PBS or with various concentrations (1, 5, and 25 mg/kg) of the CAQK peptide. After 2 weeks, blood was collected from the mice and analyzed for liver and kidney toxicity. Tests performed for liver function (ALP: alkaline phosphatase, ALT: alanine aminotransferase, TBIL total bilirubin, GGT G-glutamyl transferase) and kidney function (BUN: blood urea nitrogen). Results expressed as mean $\pm$ SD. For a more detailed safety analysis, Sprague-Dawley rats were randomized to four dose groups. CAQK or vehicle (0.9% sodium chloride injection, USP) was administered once daily by i.v. administration to male and female rats at 0 (vehicle),10, 100, or 300 mg/kg/day (Groups 1–4, respectively) for 7 consecutive days. All animals were euthanized and necropsied after blood sample collection for clinical pathology on Day 8. All animals in Groups 1–4 were evaluated for clinical observations, body weights, and clinical pathology (hematology, coagulation, and serum chemistry), gross necropsy observations and major organ weights.

## RNA sample preparation and data analysis

Adult male C57BL/6 mice with CCI were treated with CAQK or vehicle ($n = 3$/group), and cortical tissue was microdissected and flash frozen at 14 dpi. RNA was collected, and mRNA copy number was determined using a NanoString nCounter mouse neuroinflammation and neuropathology panels. Approximately 75 mg of cortical brain was isolated from sham and TBI mice (for TBI samples, the cortical tissue collected was ~5 mm of the cortical area surrounding the ipsilateral injury site). RNA was isolated using Trizol reagent (Invitrogen, 15,596–018). RNA concentration and quality were assessed using the Nanodrop 2000 spectrophotometer (Thermo Scientific). Total RNA (20 ng/µl) was run on a NanoString nCounter® system for the Mouse Neuroinflammation v1.0 panel and Mouse Neuropathology panel (NanoString Technologies, Seattle, WA) to profile RNA transcript counts for over 1400 genes and 20 housekeeping genes. Sample gene transcript counts were normalized prior to downstream analysis, and pairwise differential expression analysis was performed with NanoString's nSolver software Version 4.0. Subsets of genes displayed as heatmaps were normalized across samples as

**The paper explained**

**Problem**

Traumatic brain injury (TBI) is a major clinical problem for the civilian population and military warfighters alike due to its high prevalence and long-term disability. There are limited therapeutic agents to reduce the ensuing secondary injury following the primary impact that contributes to poor outcomes after TBI.

**Results**

In this study, we report a novel activity of a TBI-targeting tetrapeptide, CAQK, that, when administered intravenously after TBI, specifically binds to tenascin-C in the injured extracellular matrix and reduces the secondary injury around the primary injury site as evaluated by improvement in neuroinflammation, cell survival and most importantly behavioral deficit in animals with brain injury.

**Impact**

The current study suggests that CAQK has a neuroprotective activity in TBI, with the possibility of extending these results to other types of injuries of the central nervous system. The clinical impact, if effective in humans, holds the promise of substantially reducing the long-term disabilities resulting from brain injuries.

z-scores and then averaged to a single value per group before plotting with GraphPad Prism. Pathway analyses was performed using Rosalind software (https://rosalind.onramp.bio/) developed by OnRamp BioInformatics, Inc. (San Diego, CA).

## Human tissue sections

Human brain tissues with acute traumatic brain injury were obtained from Banner Sun Health Research Institute in Sun City, Arizona. Trauma 1 is a 91-year-old patient with subacute head injury, occipital scalp contusions and subgaleal blood clots, occipital and right temporal subdural hematoma and hemorrhagic contusions of the right anterior temporal pole. Trauma 2 is a patient with moderate TBI who died at age 72 in an automobile accident. The control case was obtained from the brain tissue repository maintained by the Center for Neuroscience & Regenerative Medicine (CNRM) at the Uniformed Services University of the Health Sciences (USU) in Bethesda, MD. Control case is from a 63-year-old male without any neurologic diagnosis or any signs of TBI on detailed neuropathologic evaluation. All sections are from the prefrontal cortex region.

## Graphics

The synopsis Graphics were created with BioRender.com.

## Statistical analysis

Data were tested for normality and were found to be normally distributed. Accordingly, data are presented as the mean ± SEM unless otherwise noted. Statistical differences between groups were assessed using paired and unpaired Student $t$-test, where appropriate. All the significance analysis was done using Statistica 8.0 software, using one-way ANOVA or two-tailed heteroscedastic Student's $t$-test. Values of $p < 0.05$ indicate statistical significance.

The details of the statistical tests carried out are indicated in the respective figure legends.

## Data availability

This study includes no data deposited in external repositories. The source data of this paper are collected in the BioImage Archive at the following record:

https://www.ebi.ac.uk/biostudies/studies/S-BSST2075

The source data of this paper are collected in the following database record: biostudies:S-SCDT-10_1038-S44321-025-00312-5.

## Peer review information

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

## Acknowledgements

We thank Dr Venkata Ramana Kotamraju for peptide synthesis, Guillermina Garcia at the SBP Core facility for assistance in histology and Alex Campos at the SBP Proteomics facility for proteomics analysis. We also thank Dr. Amanda Roberts at the Scripps Research Institute for help with behavioral testing done in mice. We also thank the Center for Neuroscience & Regenerative Medicine Brain Tissue Repository at the Uniformed Services University of the Health Sciences in Bethesda, MD, for sharing human brain tissues. This work was supported by the National Institutes of Health (R43NS112050 and R44NS130776-01 to APM) and the National Science Foundation (1548490 and 1660165 to SH).

## Author contributions

**Aman P Mann**: Conceptualization; Data curation; Formal analysis; Supervision; Funding acquisition; Investigation; Visualization; Methodology; Writing—original draft; Project administration; Writing—review and editing. **Sazid Hussain**: Conceptualization; Data curation; Formal analysis; Supervision; Funding acquisition; Validation; Investigation; Visualization; Methodology; Writing—original draft; Project administration; Writing—review and editing. **Pablo Scodeller**: Data curation; Investigation. **Hope N B Moore**: Data curation; Investigation; Methodology. **Elan Sherazee**: Data curation; Investigation; Methodology. **Rachel M Russo**: Supervision; Investigation; Methodology; Project administration; Writing—review and editing. **Erkki Ruoslahti**: Conceptualization; Writing—review and editing.

Source data underlying figure panels in this paper may have individual authorship assigned. Where available, figure panel/source data authorship is listed in the following database record: biostudies:S-SCDT-10_1038-S44321-025-00312-5.

## Disclosure and competing interests statement

APM, SH, PS, and ER are inventors on the patents. APM, SH and ER have ownership interest in AivoCode and are founders and officers of the company. The remaining authors declare no competing financial interests.

# Expanded View Figures

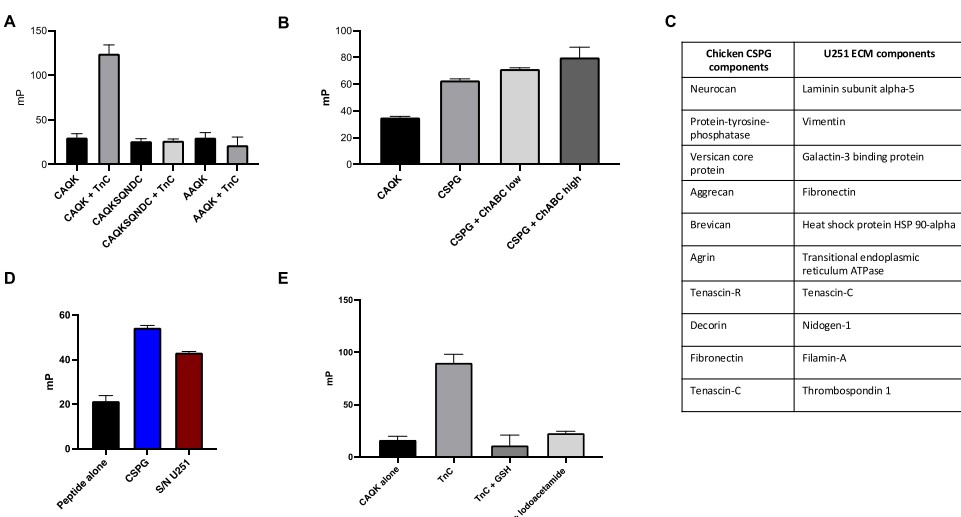

**Figure EV1. CAQK binding to CSPG components.**

(A) Fluorescence polarization (FP) measurement of CAQK binding to CSPG. FAM-CAQK (20 nM) was incubated with CSPG (200 nM) for 60 min at 37 °C. Binding with CSPG pretreated with chondroitinase ABC (chABC) at two concentrations low (5mU) and high (15mU) was compared to untreated CSPG. (B) FP assay to assess the binding of CAQK to brain ECM from different sources. FAM-labeled CAQK (20 nM) was incubated for 1 h at 37 °C with purified CSPG isolated from chicken brain (1 μM) or supernatant collected from cultured U251 human glioblastoma cell line. (C) Top hits from proteomic analysis of CSPG complex isolated from chicken brains and U251 conditioned media. (D) FP assay of binding of different peptides to TnC. FAM-labeled peptides (20 nM) were incubated with TnC (1 μM) for 1 h at 37 °C. (E) FP assay to assess the effect of the thiol group in cysteine of CAQK on binding to TnC. FAM-labeled CAQK (20 nM) was incubated with TnC (1 μM) for 1 h at 37 °C in the presence of GSH and Iodoacetamide. $n = 3$.

**A**

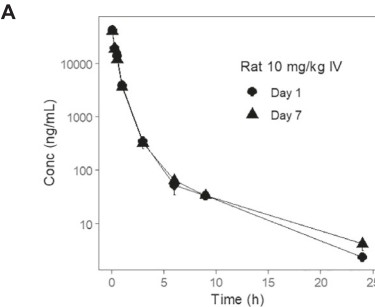

**B**

| Parameter | Unit | Day 1 | Day 7 |
|---|---|---|---|
| k10 | 1/h | 1.7 | 1.7 |
| k12 | 1/h | 0.074 | 0.090 |
| k21 | 1/h | 0.19 | 0.16 |
| t1/2Alpha | h | 0.39 | 0.38 |
| t1/2Beta | h | 3.9 | 4.6 |
| C0 | µg/mL | 30.8 | 28.9 |
| V1 | L/kg | 0.33 | 0.35 |
| CL1 | mL/min/kg | 9.3 | 10.0 |
| V2 | L/kg | 0.13 | 0.19 |
| CL2 | mL/min/kg | 0.40 | 0.52 |
| AUC 0-t | µg/mL*h | 18.0 | 16.7 |
| AUC 0-inf | µg/mL*h | 18.0 | 16.7 |
| MRT | h | 0.82 | 0.90 |
| Vss | L/kg | 0.45 | 0.54 |

**C**

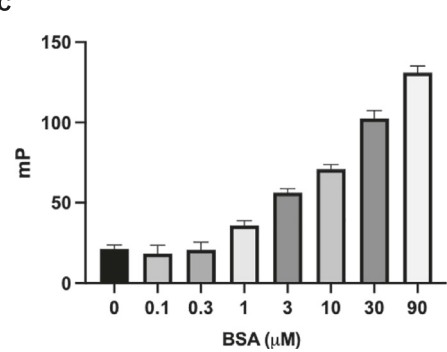

**Figure EV2. CAQK plasma clearance in rats.**

(**A**) CAQK was injected daily i.v. in healthy adult Sprague-Dawley rats at a 10 mg/kg dose for 7 days. Blood was collected at different time points at Day 1 and Day 7, and plasma was analyzed by LC/MS to analyze CAQK. Plasma concentration of CAQK was plotted. $N = 3$ per time point. Plasma clearance data were also used to calculate pharmacokinetic parameters in (**B**). (**C**) FP measurement of CAQK binding to BSA. FAM-CAQK (20 nM) was incubated with BSA at the indicated concentration for 1 h at 37 °C. $n = 3$.

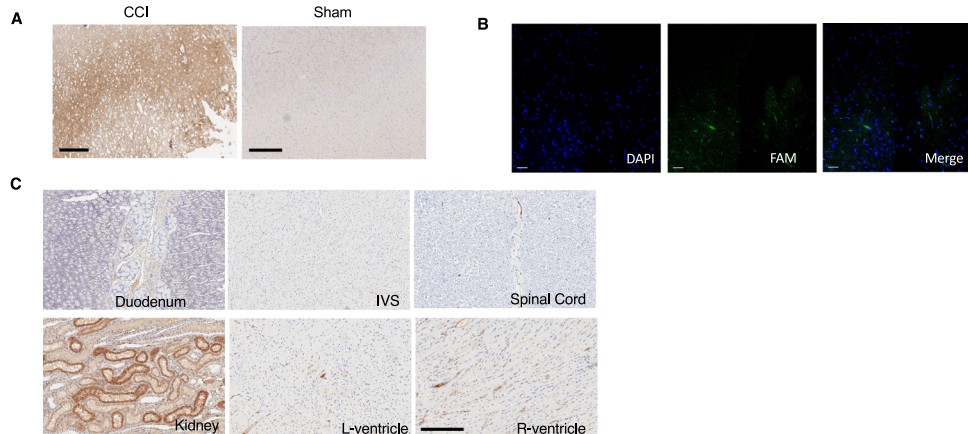

**Figure EV3.  CAQK targeting in the pig model of TBI.**

(**A**) TnC expression is upregulated in pig CCI. Immunohistochemical staining for TnC on cortical pig brain sections shows elevated tenascin-C expression in the cortex surrounding CCI brain injury compared to the cortex of a sham-injured animal. Scale bar—300 μm. (**B**) Control peptide does not accumulate in the brain of a pig with CCI. Fluorescence imaging on cortical brain sections from CCI pig injected with FAM-AAQK. Sections were immunostained with anti-FAM (AAQK; green) and nuclei (blue). Scale bar—100 μm. (**C**) CAQK accumulation in different organs in pig CCI. A male Yorkshire pig with CCI was injected with FAM-CAQK at 6 h post-injury. IHC staining of FAM label on fixed paraffin-embedded tissue sections from different organs collected 1 h after peptide injection. Scale bar, 200 μm. IVS interventricular septum.

