## [Peer Review File · EMBO Molecular Medicine]

A neuroprotective tetrapeptide for treatment of acute traumatic brain injury

Aman Mann, Sazid Hussain, Pablo Scodeller, Hope Moore, Elan Sherazee, Rachel Russo, and Erkki Ruoslahti

Corresponding author: Aman Mann (amann@aivocode.com)

Review Timeline:

Submission Date:	11th May 24
Editorial Decision:	27th Jun 24
Revision Received:	28th May 25
Editorial Decision:	18th Jul 25
Revision Received:	28th Aug 25
Accepted:	4th Sep 25

Editors: Poonam Bheda / Jingyi Hou

Transaction Report:

27th Jun 2024

Dear Dr. Mann,

Thank you for the submission of your manuscript to EMBO Molecular Medicine. We have now received feedback from the three reviewers who agreed to evaluate your manuscript. As you will see from the reports below, the referees acknowledge the interest of the study and are overall supportive of your work; however they also comment on multiple aspects of the manuscript that should be strengthened in a revision. In particular, it will be important to include additional pharmacokinetic analyses and timepoints after multiple treatments with the CAQK peptide (Reviewers 1 and 2) as well as test whether the FAM labeling affects the properties of the peptide and improve the histological evaluation of the treatment (Reviewer 2). In addition, the revised manuscript should include more and consistent controls, as well as additional behavioral and molecular tests (Reviewers 1 and 3). However, as the referees agree that the study appears to be promising therapeutically, editorially we suggested in a cross-commenting session with the reviewers that we would not require further extending the manuscript to include testing D-enantiomers and alternative routes of administration, to which the reviewers agreed.

Addressing the reviewers' concerns in full in a point-by-point response will be necessary for further considering the manuscript in our journal, and acceptance of the manuscript will entail a second round of review. EMBO Molecular Medicine encourages a single round of revision only and therefore, acceptance or rejection of the manuscript will depend on the completeness of your responses included in the next, final version of the manuscript. For this reason, and to save you from any frustrations in the end, I would strongly advise against returning an incomplete revision. If you would like to discuss further the points raised by the referees, I am available to do so via email or video. Let me know if you are interested in this option.

We are expecting your revised manuscript within three months, if you anticipate any delay, please contact us. When submitting your revised manuscript, please carefully review the instructions that follow below. We perform an initial quality control of all revised manuscripts before re-review; failure to include requested items will delay the evaluation of your revision.

We require:

- 1) A .docx formatted version of the manuscript text (including legends for main figures, EV figures and tables). Please make sure that the changes are highlighted to be clearly visible.
- 2) Individual production quality figure files as .eps, .tif, .jpg (one file per figure). For guidance, download the 'Figure Guide PDF' (<https://www.embopress.org/page/journal/17574684/authorguide#figureformat>).
- 3) At EMBO Press we ask authors to provide source data for the main figures. Our source data coordinator will contact you to discuss which figure panels we would need source data for and will also provide you with helpful tips on how to upload and organize the files.
- 4) A .docx formatted letter INCLUDING the reviewers' reports and your detailed point-by-point responses to their comments. As part of the EMBO Press transparent editorial process, the point-by-point response is part of the Review Process File (RPF), which will be published alongside your paper.
- 5) A complete author checklist, which you can download from our author guidelines (<https://www.embopress.org/page/journal/17574684/authorguide#submissionofrevisions>). Please insert information in the checklist that is also reflected in the manuscript. The completed author checklist will also be part of the RPF.
- 6) Please note that all corresponding authors are required to supply an ORCID ID for their name upon submission of a revised manuscript.
- 7) It is mandatory to include a 'Data Availability' section after the Materials and Methods. Before submitting your revision, primary datasets produced in this study need to be deposited in an appropriate public database, and the accession numbers and database listed under 'Data Availability'. Please remember to provide a reviewer password if the datasets are not yet public (see <https://www.embopress.org/page/journal/17574684/authorguide#dataavailability>).

In case you have no data that requires deposition in a public database, please state so in this section. Note that the Data Availability Section is restricted to new primary data that are part of this study.
This study includes no data deposited in external repositories.

- 8) All Materials and Methods need to be described in the main text using our 'Structured Methods' format, which is required for all research articles. According to this format, the Methods section includes a Reagents and Tools Table (listing key reagents, experimental models, software and relevant equipment and including their sources and relevant identifiers) followed by a

Methods and

Protocols section describing the methods using a step-by-step protocol format. The aim is to facilitate adoption of the methodologies across labs. More information on how to adhere to this format as well as a downloadable template (.docx) for the Reagents and Tools Table can be found in our author guidelines:

<https://www.embopress.org/page/journal/17574684/authorguide#structuredmethods>

9) For data quantification: please specify the name of the statistical test used to generate error bars and P values, the number (n) of independent experiments (specify technical or biological replicates) underlying each data point and the test used to calculate p-values in each figure legend. The figure legends should contain a basic description of n, P and the test applied. Graphs must include a description of the bars and the error bars (s.d., s.e.m.). Please provide exact p values.

10) Our journal encourages inclusion of *data citations in the reference list* to directly cite datasets that were re-used and obtained from public databases. Data citations in the article text are distinct from normal bibliographical citations and should directly link to the database records from which the data can be accessed. In the main text, data citations are formatted as follows: "Data ref: Smith et al, 2001" or "Data ref: NCBI Sequence Read Archive PRJNA342805, 2017". In the Reference list, data citations must be labeled with "[DATASET]". A data reference must provide the database name, accession number/identifiers and a resolvable link to the landing page from which the data can be accessed at the end of the reference. Further instructions are available at .

11) We replaced Supplementary Information with Expanded View (EV) Figures and Tables that are collapsible/expandable online. A maximum of 5 EV Figures can be typeset. EV Figures should be cited as 'Figure EV1, Figure EV2" etc... in the text and their respective legends should be included in the main text after the legends of regular figures.

12) The paper explained: EMBO Molecular Medicine articles are accompanied by a summary of the articles to emphasize the major findings in the paper and their medical implications for the non-specialist reader. Please provide a draft summary of your article highlighting

- the medical issue you are addressing,

- the results obtained and

- their clinical impact.

13) For more information: There is space at the end of each article to list relevant web links for further consultation by our readers. Could you identify some relevant ones and provide such information as well? Some examples are patient associations, relevant databases, OMIM/proteins/genes links, author's websites, etc...

14) Author contributions: CRediT has replaced the traditional author contributions section because it offers a systematic machine readable author contributions format that allows for more effective research assessment. Please remove the Authors Contributions from the manuscript and use the free text boxes beneath each contributing author's name in our system to add specific details on the author's contribution. More information is available in our guide to authors.

15) Disclosure statement and competing interests: We updated our journal's competing interests policy in January 2022 and request authors to consider both actual and perceived competing interests. Please review the policy

<https://www.embopress.org/competing-interests> and update your competing interests if necessary.

16) Every published paper now includes a 'Synopsis' to further enhance discoverability. Synopses are displayed on the journal webpage and are freely accessible to all readers. They include a short stand first (maximum of 300 characters, including space) as well as 2-5 one-sentences bullet points that summarizes the paper. Please write the bullet points to summarize the key NEW findings. They should be designed to be complementary to the abstract - i.e. not repeat the same text. We encourage inclusion of key acronyms and quantitative information (maximum of 30 words / bullet point). Please use the passive voice. Please attach these in a separate file or send them by email, we will incorporate them accordingly.

Please also suggest a visual abstract to illustrate your article as a jpeg file 550 px wide x 300-600 px high.

Share synopsis text and image, as well as eTOC:

Please note that these would be the final versions and changes during proofing are usually not allowed

17) As part of the EMBO Publications transparent editorial process initiative (see our policy here:

https://www.embopress.org/transparent-process#Review_Process), EMBO Molecular Medicine will publish online a Peer Review File (PRF) to accompany accepted manuscripts.

In the event of acceptance, this file will be published in conjunction with your paper and will include the anonymous referee reports, your point-by-point response and all pertinent correspondence relating to the manuscript. Let us know whether you agree with the publication of the PRF and as here, if you want to remove or not any figures from it prior to publication.

I look forward to receiving your revised manuscript.

Yours sincerely,

Poonam Bheda

Poonam Bheda, PhD
Scientific Editor
EMBO Molecular Medicine

***** Reviewer's comments *****

Referee #1 (Comments on Novelty/Model System for Author):

The study presents a novel therapeutic approach using the tetrapeptide CAQK for the treatment of acute traumatic brain injury (TBI). The findings are promising and contribute significantly to the field of neurotrauma research.

Referee #1 (Remarks for Author):

Comments:

I have carefully reviewed the manuscript titled "A neuroprotective tetrapeptide for treatment of acute traumatic brain injury" by Aman P. Mann et al. The manuscript by Mann et al. presents valuable findings that could have significant implications for the treatment of TBI. After addressing the aforementioned points, I believe the study will be well-positioned for publication. The authors are encouraged to revise the manuscript accordingly:

Question

1. The study presents a novel therapeutic approach using the tetrapeptide CAQK for the treatment of acute traumatic brain injury (TBI). The findings are promising and contribute significantly to the field of neurotrauma research. However, the manuscript would benefit from a more detailed discussion on the potential mechanisms of action of CAQK. Specifically, the role of Tenascin C (TnC) in mediating the neuroprotective effects of CAQK requires further elucidation.
2. The pharmacokinetic analysis of CAQK in mice is well-presented (Figure 2). However, it would be advantageous to include a comparison of the biodistribution of CAQK in both normal and TBI mouse models to better understand its targeting specificity.
3. The manuscript reports a significant reduction in lesion size and improved functional outcomes following CAQK treatment. While these results are compelling, the study would be strengthened by including a placebo control group to account for potential confounding variables.
4. The behavioral testing data (Figure 6) indicate functional improvements in treated mice. However, it would be helpful to include additional behavioral tests to provide a more comprehensive assessment of cognitive and motor function recovery.
5. In the Introduction, the authors mention the prevalence and impact of TBI. Adding a brief overview of current treatment options and their limitations would provide useful context for the study.

6. There are a few instances where the manuscript could benefit from improved clarity and conciseness. For example, the second paragraph of the Discussion section could be rephrased for better readability.
7. Figure 1 could be improved by providing a higher resolution image to clearly show the fluorescence polarization data.
8. The manuscript would benefit from a more thorough proofreading to correct minor typographical errors and improve the overall quality of the text.

Referee #2 (Comments on Novelty/Model System for Author):

Please see my comments below

Referee #2 (Remarks for Author):

The tetrapeptide CAQK (cysteine-alanine-glutamine-lysine) has obtained a significant interest in the field of neurotherapeutics, particularly for its potential applications in treating central nervous system (CNS) injuries such as traumatic brain injury (TBI) and spinal cord injury.

The manuscript by Mann et al. presents significant advances in understanding the neuroprotective effects of the tetrapeptide CAQK for traumatic brain injury (TBI).

While the study offers promising results, several critical points require further consideration and investigation to fully establish the therapeutic potential of CAQK.

CAQK was administered intravenously administrated daily for one week starting 6h post-TBI. PK was studied after a single administration only. The study needs more data on the pharmacokinetics and dynamics of repeated dosing. It is crucial to analyze how multiple doses affect the peptide's distribution, efficacy, and potential side effects over time. Additionally, the capacity of the peptide to trigger an immunological response with repeated administration should be thoroughly investigated. Immunogenicity is a critical factor in the long-term application of peptide-based therapies.

To address potential immunogenicity and improve the peptide's stability and permanence in the brain, the authors should consider evaluating an all-D enantiomer of the peptide. D-amino acid peptides are known for their resistance to enzymatic degradation, which could enhance the therapeutic duration and reduce the frequency of administration. The manuscript does not discuss whether such an enantiomer was considered or tested. Incorporating this analysis could provide valuable insights into optimizing the peptide's pharmacological properties.

The current study utilizes intravenous administration of the peptide. While effective, there may be better methods than this method in the milder cases. Exploring alternative routes, such as intranasal delivery, could simplify the therapy and improve patient compliance. Intranasal administration offers a non-invasive method directly targeting the central nervous system, potentially increasing the peptide's efficiency and reducing systemic exposure. The authors should investigate this route as a viable alternative to intravenous injections.

The study employs a FAM-labeled peptide to determine the distribution and binding properties of CAQK. However, it is essential to consider whether the FAM label alters the physicochemical properties of the peptide, potentially affecting its behaviour in biological systems. The authors should provide evidence or discuss the extent to which the FAM-labeled peptide retains the same characteristics as the original tetrapeptide. Furthermore, advanced imaging techniques, such as MALDI imaging, could offer more precise and direct insights into the peptide's distribution and binding properties without the potential confounding effects of a fluorescent label.

The histological evaluation of neuroinflammation needs to be deeper. A larger overview showing the areas used for quantification needs to be provided. Given the effect on the memory test, the authors should also evaluate neuroinflammatory markers in the hippocampus.

In rodents, after TBI, blood GFAP levels peak within the first 24 hours following injury. While the use of GFAP levels at 24 hours as a surrogate marker of treatment efficacy has been widely reported in rodents after TBI, an effect on GFAP levels 14 days after injury is quite unusual, as GFAP levels at this time point are close to naïve values. How do the authors explain the absence of differences 24 hours after injury? Furthermore, for some CAQK-treated mice, GFAP levels at 24 hours were higher. This could indicate a worsening brain injury in a subset of treated mice. A correlation between lesion volume and 24-hour GFAP levels in the CAQK-treated group could help clarify this.

It has recently been shown that blood neurofilament light levels (NfL) were reduced after CCI in mice treated with A β 1-6A2V(D) (PMID: 37198260). Moreover, given that blood NfL shows a delayed peak compared to GFAP after TBI and the authors' finding that CAQK reduces GFAP at 14 days but not at 24 hours, the manuscript would benefit from including an assessment of plasma NfL. This addition would provide a more comprehensive evaluation of potential biomarkers' translational value as a secondary endpoint for clinical studies.

Considering that TnC has been shown to affect the proliferation, migration, survival, and differentiation of oligodendrocytes, as

correctly reported by the Authors, a deeper exploration of the effect of CAQK on this cell type should be provided across species (human, pig and mice)

In conclusion, while the manuscript presents a compelling case for the neuroprotective effects of CAQK, addressing these critical points would strengthen the study's impact and applicability. Repeated dosing studies, evaluation of all-D enantiomer peptides, exploration of alternative administration routes, and confirmation of the FAM-labeled peptide's equivalence to the original are necessary to fully validate and optimize CAQK as a therapeutic agent for TBI.

Referee #3 (Remarks for Author):

Aman P. Mann and colleagues investigated CAQK may have therapeutic applications in TBI. Overall, the paper is well-written, and the results are compelling.

Specific comments:

1. The ARRIVE criteria for reporting animal experiments are not fully fulfilled, e.g. the number of the license allowing to perform the described experiments is not given. Please use and provide an ARRIVE criteria checklist.
2. Was all data normally distributed?
3. Methods for quantification of the immunofluorescence are needed.
4. If possible, please provide a low magnification fluorescence image , e.g. Fig 1 C and Fig 4E. In addition, it is best to have a consistent style for all scale bar in the figures.
5. In Fig 4E, IBA1 is the marker of microglia, M1 and M2 typing tests will be more convincing.
6. I was confused that the comparisons in Fig 6 A and C (CAQK vs Vehicle) were different in in Fig 6 B (Vehicle vs Naive).
7. The writing level needs further improvement, such as capitalization matters (e.g."Fig.1." in the figure legend). Besides, the group names were missing in Fig 4E.

Response to reviewers:**Referee #1** (Comments on Novelty/Model System for Author):

The study presents a novel therapeutic approach using the tetrapeptide CAQK for the treatment of acute traumatic brain injury (TBI). The findings are promising and contribute significantly to the field of neurotrauma research.

Referee #1 (Remarks for Author):

Comments:

I have carefully reviewed the manuscript titled "A neuroprotective tetrapeptide for treatment of acute traumatic brain injury" by Aman P. Mann et al. The manuscript by Mann et al. presents valuable findings that could have significant implications for the treatment of TBI. After addressing the aforementioned points, I believe the study will be well-positioned for publication. The authors are encouraged to revise the manuscript accordingly:

1. The study presents a novel therapeutic approach using the tetrapeptide CAQK for the treatment of acute traumatic brain injury (TBI). The findings are promising and contribute significantly to the field of neurotrauma research. However, the manuscript would benefit from a more detailed discussion on the potential mechanisms of action of CAQK. Specifically, the role of Tenascin C (TnC) in mediating the neuroprotective effects of CAQK requires further elucidation.
 - We have added a more detailed discussion (Discussion - Paragraph 5) about the potential mechanisms of action of CAQK in TBI.
2. The pharmacokinetic analysis of CAQK in mice is well-presented (Figure 2). However, it would be advantageous to include a comparison of the biodistribution of CAQK in both normal and TBI mouse models to better understand its targeting specificity.
 - We have previously conducted a biodistribution study that showed that i.v. administered CAQK specifically localizes in TBI brain compared to uninjured brain or other organs, except for the kidney which is the route of peptide clearance (Mann et. al. Nature Communications 2016). In that publication, we also showed that systemically circulating CAQK does not target a liver perforation injury and is therefore specific to injuries of the central nervous system. In our current study, we confirmed the specific accumulation of the peptide to injured brain in pig model of TBI and evaluated peptide accumulation in other organs, which were negative except for kidneys. We have now added this information in the results section.

3. The manuscript reports a significant reduction in lesion size and improved functional outcomes following CAQK treatment. While these results are compelling, the study would be strengthened by including a placebo control group to account for potential confounding variables.
 - We included a vehicle control group in our efficacy study which was just saline injection which can be considered a placebo control group.
4. The behavioral testing data (Figure 6) indicate functional improvements in treated mice. However, it would be helpful to include additional behavioral tests to provide a more comprehensive assessment of cognitive and motor function recovery.
 - We included a battery of behavioral tests in our study to assess different aspects of motor coordination and cognitive function. Expanding this panel would be outside the scope of this first description of CAQK activity.
5. In the Introduction, the authors mention the prevalence and impact of TBI. Adding a brief overview of current treatment options and their limitations would provide useful context for the study.
 - We have added that context in the introduction (Paragraph 1)
6. There are a few instances where the manuscript could benefit from improved clarity and conciseness. For example, the second paragraph of the Discussion section could be rephrased for better readability.
 - We have reviewed the entire manuscript and rewritten much of it as described above.
7. Figure 1 could be improved by providing a higher resolution image to clearly show the fluorescence polarization data.
 - We have replaced the Fig. 1A with a higher-resolution image of the fluorescence polarization data.
8. The manuscript would benefit from a more thorough proofreading to correct minor typographical errors and improve the overall quality of the text.
 - The three main authors have proofread the final version of the manuscript.

Referee #2

1. PK was studied after a single administration only. The study needs more data on the pharmacokinetics and dynamics of repeated dosing. It is crucial to analyze how multiple doses affect the peptide's distribution, efficacy, and potential side effects over time. Additionally, the capacity of the peptide to trigger an immunological response with repeated administration should be thoroughly investigated. Immunogenicity is a critical factor in the long-term application of peptide-based therapies.
 - This, and several other comments by this reviewer ask for data that are typically generated later in the process of developing a drug candidate

than an initial report of a new compound. However, we have conducted a toxicity and PK study on 7-day repeat dosing. We did not observe any toxicity (no effect on body weight or microscopic or gross pathology) at a dose as high as 300 mg/kg/day of repeat IV dosing. The pK profile of the peptide was unaltered after 7-day repeat dosing. We have included this data in the revised manuscript (Fig. 2). Short peptides, i.e. under 8 aa, are usually reported as non-immunogenic unless coupled to a protein carrier.

2. To address potential immunogenicity and improve the peptide's stability and permanence in the brain, the authors should consider evaluating an all-D enantiomer of the peptide. D-amino acid peptides are known for their resistance to enzymatic degradation, which could enhance the therapeutic duration and reduce the frequency of administration. The manuscript does not discuss whether such an enantiomer was considered or tested. Incorporating this analysis could provide valuable insights into optimizing the peptide's pharmacological properties.
 - We have not tested the D-amino acid peptides as it was out of scope of the current study.
3. The current study utilizes intravenous administration of the peptide. While effective, there may be better methods than this method in the milder cases. Exploring alternative routes, such as intranasal delivery, could simplify the therapy and improve patient compliance. Intranasal administration offers a non-invasive method directly targeting the central nervous system, potentially increasing the peptide's efficiency and reducing systemic exposure. The authors should investigate this route as a viable alternative to intravenous injections.
 - The eventual goal of this therapy is to be a treatment for a patient with moderate to severe brain injury. These patients are generally treated at trauma centers or emergency rooms and are already hooked up to an IV line. Therefore, IV dosing of CAQK was considered as the most effective route of administration which could easily be added to current workflow of clinical management of TBI patients. We have included this reasoning in the discussion.
4. The study employs a FAM-labeled peptide to determine the distribution and binding properties of CAQK. However, it is essential to consider whether the FAM label alters the physicochemical properties of the peptide, potentially affecting its behaviour in biological systems. The authors should provide evidence or discuss the extent to which the FAM-labeled peptide retains the same characteristics as the original tetrapeptide. Furthermore, advanced imaging techniques, such as MALDI imaging, could offer more precise and direct insights into the peptide's distribution and binding properties without the potential confounding effects of a fluorescent label.
 - The FAM label on the peptide was only used for in vitro binding and initial targeting studies in vivo. Unlabeled CAQK was used for the PK and brain accumulation studies and the peptide was analyzed by mass spectroscopy.

These studies confirmed that the unlabeled peptide closely matched the brain accumulation observed with the FAM-labeled version of CAQK.

5. The histological evaluation of neuroinflammation needs to be deeper. A larger overview showing the areas used for quantification needs to be provided. Given the effect on the memory test, the authors should also evaluate neuroinflammatory markers in the hippocampus.
 - We have added the areas used for quantification and added neuroinflammatory marker staining in the hippocampus (Fig. 4).

6. In rodents, after TBI, blood GFAP levels peak within the first 24 hours following injury. While the use of GFAP levels at 24 hours as a surrogate marker of treatment efficacy has been widely reported in rodents after TBI, an effect on GFAP levels 14 days after injury is quite unusual, as GFAP levels at this time point are close to naïve values. How do the authors explain the absence of differences 24 hours after injury? Furthermore, for some CAQK-treated mice, GFAP levels at 24 hours were higher. This could indicate a worsening brain injury in a subset of treated mice. A correlation between lesion volume and 24-hour GFAP levels in the CAQK-treated group could help clarify this.
 - Blood GFAP levels peak at early time points (around 24 hours) and strongly correlate with TBI severity. Following the primary injury, GFAP levels can continue to stay elevated because of ensuing secondary injury and continuing neuronal loss. Literature shows elevated biomarker levels including GFAP in some TBI patients up to 5 years after injury (*Manktelow A, et al Brain, 2022*). In our TBI experiments, GFAP levels at 24 hours were indicative of the extent of TBI severity in mice. There is no statistical difference in GFAP levels at 24 hours between CAQK and vehicle treated groups. It is probably too early to expect any significant improvement of the injury (and lower GFAP) after 24 hours and only one CAQK injection.

7. It has recently been shown that blood neurofilament light levels (NfL) were reduced after CCI in mice treated with A β 1-6A2V(D) (PMID: 37198260). Moreover, given that blood NfL shows a delayed peak compared to GFAP after TBI and the authors' finding that CAQK reduces GFAP at 14 days but not at 24 hours, the manuscript would benefit from including an assessment of plasma NfL. This addition would provide a more comprehensive evaluation of potential biomarkers' translational value as a secondary endpoint for clinical studies.
 - We have now tested NF-L levels at 24 hour and 14 days after TBI. We saw a slight decrease in NF-L on CAQK treatment at day 14, however, it was not statistically significant. We have added that information in Appendix Fig. 1.

8. Considering that TnC has been shown to affect the proliferation, migration, survival, and differentiation of oligodendrocytes, as correctly reported by the Authors, a deeper exploration of the effect of CAQK on this cell type should be

provided across species (human, pig and mice)

- Deeper understanding of the CAQK activity on oligodendrocytes derived from different species would be outside the scope of this initial study

Referee #3 (Remarks for Author):

Aman P. Mann and colleagues investigated CAQK may have therapeutic applications in TBI. Overall, the paper is well-written, and the results are compelling.

Specific comments:

1. The ARRIVE criteria for reporting animal experiments are not fully fulfilled, e.g. the number of the license allowing to perform the described experiments is not given.

Please use and provide an ARRIVE criteria checklist.

- The license and protocol information for the animal studies have been added to the methods section (Paragraph 1 and 10). We have also added the ARRIVE checklist.

2. Was all data normally distributed?

- Yes, the data was normally distributed

3. Methods for quantification of the immunofluorescence are needed.

- We have now added details of the method for quantification of immunofluorescence to the methods section (Paragraph 7).

4. If possible, please provide a low magnification fluorescence image, e.g. Fig 1 C and Fig 4E. In addition, it is best to have a consistent style for all scale bar in the figures.

- We have added low magnification images to the manuscript and reformatted the scale bars to be consistent

5. In Fig 4E, IBA1 is the marker of microglia, M1 and M2 typing tests will be more convincing.

- We agree Iba1 is a pan-marker for microglia. Instead of staining for M1 and M2 specific markers, we analyzed RNA transcriptomic data from TBI mice after treatment and compared with vehicle treated group to analyze expression of M1 markers of microglia (Fig. 5E). We noticed that a number of M1 specific markers were downregulated after CAQK treatment.

6. I was confused that the comparison in Fig 6 A and C (CAQK vs Vehicle) were different in in Fig 6 B (Vehicle vs Naive).

- We have added comparisons of CAQK vs vehicle to the other plots in Fig. 6 to be consistent with other charts

7. The writing level needs further improvement, such as capitalization matters (e.g. "Fig.1." in the figure legend). Besides, the group names were missing in Fig 4E.

- The group names for the figures were described in the figure 4 legend, but we have now added them in the figure for clarity. We have also proofread the manuscript.

18th Jul 2025

Dear Dr. Mann,

Thank you for submitting the revised version of your manuscript to EMBO Molecular Medicine. We have now received reports from the three referees who re-evaluated your work. As the original Reviewer #2 was unavailable for this round of review, we invited a new reviewer (Reviewer #4) to assess your responses to Reviewer #2's comments.

As you will see, Reviewer #3 is satisfied with the revisions made. Reviewer #4 considers your responses overall appropriate and acknowledges that some of the original concerns are beyond the scope of the current manuscript. However, this reviewer still raises a few minor points and recommends that the issues originally raised by Reviewer #2 be discussed in the Discussion section.

With respect to the comments from Reviewer #1, while we would not request you to address them experimentally at this stage, we kindly ask you to respond to each point and discuss them as future direction. Please also provide a clearer scientific explanation in your point-by-point response for not including the additional behavioral tests requested by Reviewer #1 during the initial round of review.

On a more editorial level, please do the following:

1. Appendix: the red font should be removed, the nomenclature for the tables should be corrected to Appendix Table S1 etc.
2. Data availability section:
 - Proteomics datasets generated in this study need to be deposited to an appropriate public database. The accession numbers and database need to be listed within this section.
3. Add missing figure callouts for Fig 2B, panels A,B,C of Fig 3, all panels of Fig 6 and Appendix Figure S2.
4. BioRender attribution: remove the BioRender mention from the Acknowledgment and add a dedicated section in the Methods using the following format:
Graphics: (some of the... OR Figure #... OR synopsis) Graphics were created with BioRender.com.
5. Author Checklist: complete the author checklist by adding corresponding author names, journal name and manuscript number.
6. Figure 3A and EV3B need to be remade with higher resolution versions. Please use the original captured 16bit tiff image to prevent pixelation.
7. Figure 4E - The Figure legend details the different magnifications of the left and right side of the figure. This should be made more clear in the figure itself.
8. The ARRIVE Checklist should be removed.
9. Please address the following issues related to figure legends:
 - Please note that legend for sub-figure 4G is not provided, kindly rectify this.
 - Please define the annotated p values ** as well as provide the exact p-values for the same in the legend of figure 1E, as appropriate.
 - Please note that the exact p values are not provided in the legends of figures 1A, B, C, D, F; 2C; 4C, D, E, F; 6A-C.
 - Please indicate the statistical test used for data analysis in the legends of figures 1E, 4G; 5A, D; S2 A, B.
 - Please note that the box plots need to be defined in terms of minima, maxima, centre, bounds of box and whiskers, and percentile in the legends of figures 4F, G; S1
 - Please note that information related to n is missing in the legends of figures 1B-F; 4D, E, G; 6C, EV1 A, B, D, E; EV2 C

I look forward to seeing a revised form of your manuscript as soon as possible.

Kind regards,
Jingyi

Jingyi Hou
Senior Editor

*** Instructions to submit your revised manuscript ***

***** Reviewer's comments *****

Referee #1 (Remarks for Author):

The authors have partially responded to my review comments. It is recommended that they perform MRI to assess edema status after CAQK treatment, and additionally examine vascular conditions. These tests can better demonstrate the favorable therapeutic effects of this small-molecule peptide.

Referee #4 (Remarks for Author):

I was asked to comment on the authors' response to comments of reviewer #2.

Reviewer 2 asked for more data on the pharmacokinetics and dynamics of repeated dosing of the CAQK peptide. I agree with the authors that this request is beyond the scope of this manuscript. However, I appreciate that the authors added experiments showing the pK profile of the peptide at repeated dosing 7days after injury.

Reviewer 2 asked for evaluating an all-D enantiomer of the peptide. I agree with the authors that this is beyond the scope of the manuscript, as authors are characterizing here the usage/applicability of a specific peptide (CAQK).

Reviewer 2 asked for another route of application of the CAQK peptide. I agree with the authors that this is beyond the scope of the manuscript.

For 1.-3. I would suggest, that authors discuss the mentioned important issues raised by Reviewer 2 in the discussion section. E.g. they could mention that future studies are required to identify the ideal interval of repetitive dosing of the peptide after injury, the evaluation of all-D enantiomers of the peptide as a potential improvement of the peptide as well as the different routes of application possible.

Reviewer 2 asked to discuss the extent to which the FAM-labeled peptide retains the same characteristics as the original tetrapeptide. The authors could sufficiently respond to this request.

Reviewer 2 asked for histological evaluation of neuroinflammation in the hippocampus, which was provided by the authors in Fig. 4 of the revised manuscript.

Reviewer 2 asked for a correlation between lesion volume and 24- hour GFAP levels in the CAQK-treated group. Authors provided this information. However, I would suggest that the authors incorporate their response explaining GFAP blood level peaks in both groups in the discussion section of the manuscript.

Reviewer 2 asked for evaluation of another biomarker (Nfl) in animals treated with the peptide and vehicle after TBI. Authors provided this information in Appendix Fig. 1.

Reviewer 2 asked for a deeper exploration of the effect of CAQK on oligodendrocytes. I agree with the authors that this request is beyond the scope of the manuscript. However, the authors already mention in the results section that TnC is an extracellular matrix protein that has been shown to affect the proliferation, migration, survival, and differentiation of cells in the oligodendrocyte lineage. Thus, one would at least expect that authors come back to this point in the discussion and hint at further studies or remove this part from the results section.

Finally, I would also ask authors to include individual values in all graph bars presented in the figures. In some graphs (e.g. Fig. 4E) individual values are presented. However, this should be consistent for all figures and is particular of interest for the behavioral data in Fig. 6.

Response to reviewers:**Referee #1**

The authors have partially responded to my review comments. It is recommended that they perform MRI to assess edema status after CAQK treatment and additionally examine vascular conditions. These tests can better demonstrate the favorable therapeutic effects of this small-molecule peptide.

- We do not have access to imaging facility to conduct MRI for this initial study describing CAQK activity. Based on the results described herein, a follow up study to test CAQK in pig model of TBI is planned which includes MRI.
- We included a battery of behavioral tests in our study to assess different aspects of motor coordination and cognitive function. Expanding this panel would be outside the scope of this first description of CAQK activity. Furthermore, mice brain differs from human brain in that it is lissencephalic, whereas the human brain and the brain of many other mammals is gyrencephalic. Our mouse data show as much as can be accomplished in mice and amply justify extending the studies to animals with a brain more similar as the human brain. Studies to validate these results in a pig model of TBI are planned.

Referee #4

Issues originally raised by Reviewer #2 have been added to the Discussion section of the manuscript.

4th Sep 2025

Dear Dr. Mann,

We are pleased to inform you that your manuscript is accepted for publication and is now being sent to our publisher to be included in the next available issue of EMBO Molecular Medicine.

Yours sincerely,

Jingyi Hou
Senior Editor
EMBO Molecular Medicine
